# Contrastive Learning Can Find An Optimal Basis For Approximately View-Invariant Functions

**Daniel D. Johnson**[1,2]**, Ayoub El Hanchi**[1]**, Chris J. Maddison**[1]
[1]University of Toronto, [2]Google Research
{ddjohnson, aelhan, cmaddis}@cs.toronto.edu

## Abstract

Contrastive learning is a powerful framework for learning self-supervised representations that generalize well to downstream supervised tasks. We show that multiple existing contrastive learning methods can be reinterpreted as learning kernel functions that approximate a fixed *positive-pair kernel*. We then prove that a simple representation obtained by combining this kernel with PCA provably minimizes the worst-case approximation error of linear predictors, under a straightforward assumption that positive pairs have similar labels. Our analysis is based on a decomposition of the target function in terms of the eigenfunctions of a positive-pair Markov chain, and a surprising equivalence between these eigenfunctions and the output of Kernel PCA. We give generalization bounds for downstream linear prediction using our Kernel PCA representation, and show empirically on a set of synthetic tasks that applying Kernel PCA to contrastive learning models can indeed approximately recover the Markov chain eigenfunctions, although the accuracy depends on the kernel parameterization as well as on the augmentation strength.

## 1 Introduction

When using a contrastive learning method such as SimCLR (Chen et al., 2020a) for representation learning, the first step is to specify the distribution of original examples $z \sim p(Z)$ within some space $\mathcal{Z}$ along with a sampler of augmented views $p(A|Z = z)$ over a potentially different space $\mathcal{A}$. [1] For example, $p(Z)$ might represent a dataset of natural images, and $p(A|Z)$ a random transformation that applies random scaling and color shifts. Contrastive learning then consists of finding a parameterized mapping (such as a neural network) which maps multiple views of the same image (e.g. draws from $a_1, a_2 \sim p(A|Z = z)$ for a fixed $z$) close together, and unrelated views far apart. This mapping can then be used to define a representation which is useful for downstream supervised learning.

The success of these representations have led to a variety of theoretical analyses of contrastive learning, including analyses based on conditional independence within latent classes (Saunshi et al., 2019), alignment of hyperspherical embeddings (Wang & Isola, 2020), conditional independence structure with landmark embeddings (Tosh et al., 2021), and spectral analysis of an augmentation graph (HaoChen et al., 2021). Each of these analyses is based on a single choice of contrastive learning objective. In this work, we go further by integrating multiple popular contrastive learning methods into a single framework, and showing that it can be used to build *minimax-optimal representations* under a straightforward assumption about similarity of labels between positive pairs.

Common wisdom for choosing the augmentation distribution $p(A|Z)$ is that it should remove irrelevant information from $Z$ while preserving information necessary to predict the eventual downstream label $Y$; for instance, augmentations might be chosen to be random crops or color shifts that affect the semantic content of an image as little as possible (Chen et al., 2020a). The goal of representation learning is then to find a representation with which we can form good estimates of $Y$ using only a few labeled examples. In particular, we focus on approximating a target function $g : \mathcal{A} \to \mathbb{R}^n$ for which $g(a)$ represents the "best guess" of $Y$ based on $a$. For regression tasks, we might be interested in a

---

[1]We focus on finite but arbitrarily large $\mathcal{Z}$ and $\mathcal{A}$, e.g. the set of 8-bit 32x32 images, and allow $\mathcal{Z} \neq \mathcal{A}$.

target function of the form $g(a) = \mathbb{E}[Y|A = a]$. For classification tasks, if $Y$ is represented as a one-hot vector, we might be interested in estimating the probability of each class, again taking the form $g(a) = \mathbb{E}[Y|A = a]$, or the most likely label, taking the form $g(a) = \text{argmax}_y\, p(Y = y|A = a)$. In either case, we are interested in constructing a representation for which $g$ can be estimated well using only a small number of labeled augmentations $(a_i, y_i)$.[2] Since we usually do not have access to the downstream supervised learning task when learning our representation, our goal is to identify a representation that enables us to approximate many different "reasonable" choices of $g$. Specifically, we focus on finding a *single* representation which allows us to approximate *every* target function with a small *positive-pair discrepancy*, i.e. every $g$ satisfying the following assumption:

**Assumption 1.1** (Approximate View-Invariance). *Each target function $g : \mathcal{A} \to \mathbb{R}$ satisfies*

$$\mathbb{E}_{p_+(a_1,a_2)}\Big[\big(g(a_1) - g(a_2)\big)^2\Big] \le \varepsilon,$$

*for some fixed $\varepsilon \in [0, \infty)$, where $p_+(a_1, a_2) = \sum_z p(a_1|z)p(a_2|z)p(z)$.*

This is a fairly weak assumption, because to the extent that the distribution of augmentations preserves information about some downstream label, our best estimate of that label should not depend much on exactly which augmentation is sampled: it should be *approximately invariant* to the choice of a different augmented view of the same example. More precisely, as long as the label $Y$ is independent of the augmentation $A$ conditioned on the original example $Z$ (i.e. assuming augmentations are chosen without using the label, as is typically the case), we must have $\mathbb{E}\big[(g(A_1) - g(A_2))^2\big] \le 2\mathbb{E}\big[(g(A) - Y)^2\big]$ (see Appendix A). For simplicity, we work with scalar $g : \mathcal{A} \to \mathbb{R}$ and $Y \in \mathbb{R}$; vector-valued $Y$ can be handled by learning a sequence of scalar functions.

Our first contribution is to unify a number of previous analyses and existing techniques, drawing connections between contrastive learning, kernel decomposition, Markov chains, and Assumption 1.1. We start by showing that minimizing existing contrastive losses is equivalent to building an approximation of a particular **positive-pair kernel**, from which a finite-dimensional representation can be extracted using Kernel PCA (Schölkopf et al., 1997). We next discuss what properties a representation must have to achieve low approximation error for functions satisfying Assumption 1.1, and show that the eigenfunctions of a Markov chain over positive pairs allow us to re-express this assumption in a form that makes those properties explicit. We then prove that, surprisingly, building a Kernel PCA representation using the positive-pair kernel is *exactly equivalent* to identifying the eigenfunctions of this Markov chain, ensuring this representation has the desired properties.

Our main theoretical result is that contrastive learning methods can be used to find a *minimax-optimal* representation for linear predictors under Assumption 1.1. Specifically, for a fixed dimension, we show that taking the eigenfunctions with the largest eigenvalues yields a basis for the linear subspace of functions that minimizes the worst case quadratic approximation error across the set of functions satisfying Assumption 1.1, and further give generalization bounds for the performance of this representation for downstream supervised learning.

We conclude by studying the behavior of contrastive learning models on two synthetic tasks for which the exact positive-pair kernel is known, and investigating the extent to which the basis of eigenfunctions can be extracted from trained models. As predicted by our theory, we find that the same eigenfunctions can be recovered from multiple model parameterizations and losses, although the accuracy depends on both kernel parameterization expressiveness and augmentation strength.

## 2 CONTRASTIVE LEARNING IS SECRETLY KERNEL LEARNING

Standard contrastive learning approaches can generally be decomposed into two pieces: a **parameterized model** that takes two augmented views and assigns them a real-valued similarity, and a **contrastive loss function** that encourages the model to assign higher similarity to positive pairs than negative pairs. In particular, the InfoNCE / NT-XEnt objective proposed by Van den Oord et al. (2018) and Chen et al. (2020a) and used with the SimCLR architecture, the NT-Logistic objective also considered by Chen et al. (2020a) and theoretically analyzed by Tosh et al. (2021), and the Spectral Contrastive Loss introduced by HaoChen et al. (2021) all have this structure.

---

[2]If we have a dataset of labeled un-augmented examples $(z_i, y_i)$, we can build a dataset of labeled augmentations by sampling one or more augmentations of each example in our original dataset.

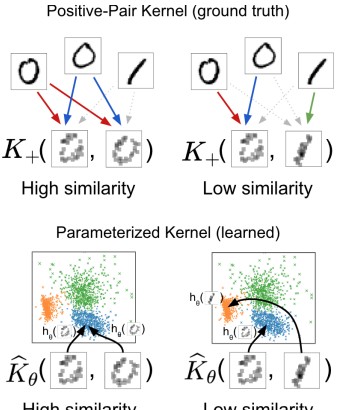

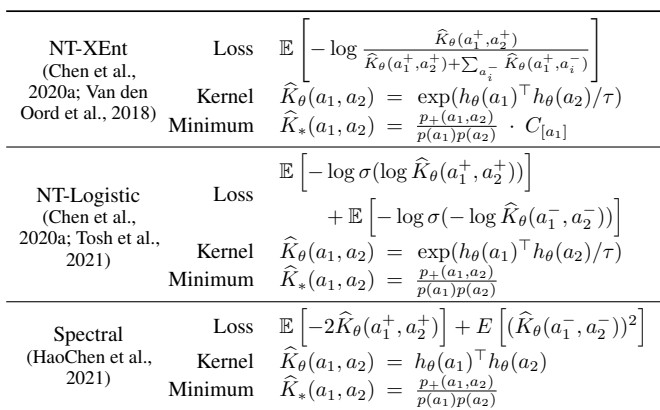

Figure 1: The positive-pair kernel $K_+$ assigns high similarity to likely positive pairs. Contrastive learning methods learn parameterized kernels $\widehat{K}_\theta$ which assign high similarity to nearby points in a learned embedding space.

Table 1: Existing contrastive learning objectives, reinterpreted as learning parameterized approximations of $K_+$. "Minimum" denotes the population minimum of the loss over all kernel functions (not necessarily representable using the shown parameterization). $C_{[a_1]}$ is a equivalence-class-dependent proportionality constant, with $C_{[a_1]} = C_{[a_2]}$ whenever $p_+(a_1, a_2) > 0$. See Appendix B for derivations and discussion of other related objectives.

The similarity between the two augmented views is commonly taken to be the dot product of outputs of a neural network, e.g. as $h_\theta(a_1)^\top h_\theta(a_2)$. However, a surprising pattern emerges if we instead interpret the *exponentiated* dot product $\exp(h_\theta(a_1)^\top h_\theta(a_2)/\tau)$ as the similarity for the NT-XEnt and NT-Logistic objectives, treating the exponential and temperature term $\tau$ as part of the model instead of part of the objective. As shown in Table 1, the three losses now share the same population minimum: the probability ratio $p_+(a_1, a_2)/p(a_1)p(a_2)$.

Intriguingly, the expressions $h_\theta(a_1)^\top h_\theta(a_2)$ and $\exp(h_\theta(a_1)^\top h_\theta(a_2)/\tau)$ both satisfy the definition of a *Mercer kernel* (also called a *positive-definite kernel*): each implicitly computes the inner product between feature vectors $\langle \phi(a_1), \phi(a_2) \rangle$ under some transformation $\phi : \mathcal{A} \to \mathbb{R}^d$ (where $d$ may be infinite). Furthermore, the probability ratio $p_+(a_1, a_2)/p(a_1)p(a_2)$ can be interpreted as a Mercer kernel as well:

**Definition 2.1.** *The **positive-pair kernel** associated with distributions $p(z)$ and $p(a|z)$ is the ratio*

$$K_+(a_1, a_2) = \frac{p_+(a_1, a_2)}{p(a_1)p(a_2)} = \langle \phi_+(a_1), \phi_+(a_2) \rangle, \tag{1}$$

*where $p_+(a_1, a_2) = \sum_z p(a_1|z)p(a_2|z)p(z)$, and $\phi_+ : \mathcal{A} \to \mathbb{R}^{|\mathcal{Z}|}$ is the transformation*

$$\phi_+(a) = \begin{bmatrix} \frac{p(a|z_1)\sqrt{p(z_1)}}{p(a)} & \frac{p(a|z_2)\sqrt{p(z_2)}}{p(a)} & \cdots & \frac{p(a|z_{|\mathcal{Z}|})\sqrt{p(z_{|\mathcal{Z}|})}}{p(a)} \end{bmatrix}^\top. \tag{2}$$

Here, the magnitude of the dot product between vectors $\phi_+(a_1)$ and $\phi_+(a_2)$ reflects the relative likelihood of data points $a_1$ and $a_2$ being drawn from a positive pair v.s. a negative pair. In particular, if $a_1$ and $a_2$ have zero probability of being a positive pair, $\phi_+(a_1)$ and $\phi_+(a_2)$ are orthogonal.

As shown in Figure 1, we can thus reinterpret the three contrastive learning methods in Table 1 as *kernel learning* methods, in that they produce parameterized positive-definite kernel functions which approximate this positive-pair kernel. By investigating properties of this kernel, we can thus hope to build a better understanding of the behavior of contrastive learning.

## 3 KERNEL PRINCIPAL COMPONENTS ARE MARKOV CHAIN EIGENFUNCTIONS

We start by investigating the geometric structure of the data under $K_+$, and how we could use Kernel PCA to build a natural representation based on this structure. We next ask what properties we would

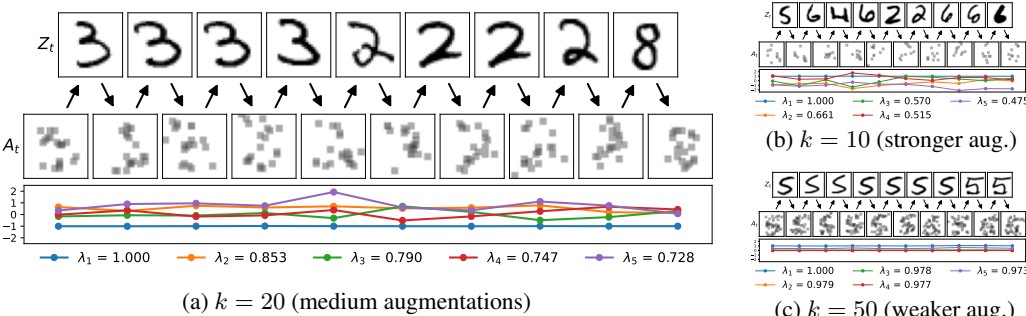

Figure 2: Samples from the positive-pair Markov chain for MNIST $k$-pixel-subsampling augmentations at three strengths ($k = 10, 20, 50$). At each step we condition on an augmentation $a_t$ (middle row) to sample an uncorrupted example $z_t$ (top row), then sample $a_{t+1}$ from $z_t$ so that $(a_t, a_{t+1})$ is a positive pair. Below, we plot the five slowest-varying eigenfunctions $f_1, \ldots, f_5$ at each step of the chain, labeled with their eigenvalues $\lambda_1, \ldots, \lambda_5$. Weaker augmentations lead to slower mixing, smoother eigenfunctions, and eigenvalues closer to 1.

want this representation to have, and use a Markov chain over positive pairs to decompose those properties in a convenient form. We then show that, surprisingly, the representation derived from $K_+$ exactly corresponds to the Markov chain decomposition and thus has precisely the desired properties.

## 3.1 SUMMARIZING $K_+$ WITH KERNEL PRINCIPAL COMPONENTS ANALYSIS

Recall that for any $a_1$ and $a_2$, we have $K_+(a_1, a_2) = \langle \phi_+(a_1), \phi_+(a_2) \rangle$ where $\phi_+$ is defined in Equation 2.1. Unfortunately, the "features" $\phi_+(a) \in \mathbb{R}^{|\mathcal{Z}|}$ are very high dimensional, being potentially as large as the cardinality of $|\mathcal{Z}|$. A natural approach for building a lower-dimensional representation is to use principal components analysis (PCA): construct the (uncentered) covariance matrix $\Sigma = \mathbb{E}_{p(a)}[\phi_+(a)\phi_+(a)^\top] \in \mathbb{R}^{|\mathcal{Z}| \times |\mathcal{Z}|}$ and then diagonalize it as $\Sigma = UDU^\top$ to determine the principal components $\{(\mathbf{u}_1, \sigma_1^2), (\mathbf{u}_2, \sigma_2^2), \ldots\}$ of our transformed data distribution $\phi_+(A)$, i.e. the directions capturing the maximum variance of $\phi_+(A)$. We can then project the transformed points $\phi_+(a)$ onto these directions to construct a sequence of "projection functions" $h_i(a) := \mathbf{u}_i^\top \phi(a)$.

Conveniently, it is possible to estimate these projection functions given access only to the kernel function $K_+(a_1, a_2) = \langle \phi_+(a_1), \phi_+(a_2) \rangle$, by using Kernel PCA (Schölkopf et al., 1997). Kernel PCA bypasses the need to estimate the covariance in feature space, directly producing the set of principal component projection functions $h_1, h_2, \ldots$ and corresponding eigenvalues $\sigma_1^2, \sigma_2^2, \ldots$, such that $h_i(a)$ measures the projection of $\phi_+(a)$ onto the $i$th eigenvector of the covariance matrix, and $\sigma_i^2$ measures the variance along that direction.

The sequence of principal component projection functions gives us a view into the geometry of our data when mapped through $\phi_+(A)$. It is thus natural to construct a $d$-dimensional representation $r : \mathcal{A} \to \mathbb{R}^d$ by taking the first $d$ such functions $r(a) = [h_1(a), h_2(a), \ldots, h_d(a)]$, and then use this for downstream learning, as is done in (kernel) principal component regression (Rosipal et al., 2001; Wibowo & Yamamoto, 2012). In practice, we can also substitute our learned kernel $\widehat{K}_\theta$ in place of $K_+$. Note that we are free to choose $d$ to trade off between the complexity of the representation and the amount of variance of $\phi_+(A)$ that we can capture.

## 3.2 DECOMPOSING INVARIANCE WITH THE POSITIVE-PAIR MARKOV CHAIN

What properties might we want this representation to have? If we wish to estimate functions $g$ satisfying Assumption 1.1, for which $\mathbb{E}_{p_+(a_1, a_2)}\left[(g(a_1) - g(a_2))^2\right]$ is small, we might hope that $\mathbb{E}_{p_+(a_1, a_2)}\left[\|r(a_1) - r(a_2)\|_2^2\right]$ is small. But this is not sufficient to ensure we can estimate $g$ with high accuracy; as an example, a constant representation is unlikely to work well. Good representation learning approaches must ensure that the learned representations are also expressive enough to approximate $g$ (for instance by using negative samples or by directly encouraging diversity as in VICReg Bardes et al. (2021)), but it is not immediately obvious what it means to be "expressive enough" if all we know about $g$ is that it satisfies Assumption 1.1.

We can build a better understanding of the quality of a representation by expanding $g$ in terms of a basis in which $\mathbb{E}_{p_+(a_1, a_2)}\big[(g(a_1) - g(a_2))^2\big]$ admits a simpler form. In particular, a convenient decomposition arises from considering the following Markov chain (shown in Figure 2): starting with an example $a_t$, sample the next example $a_{t+1}$ proportional to how likely $(a_t, a_{t+1})$ would be a positive pair, i.e. according to $p_+(a_{t+1}|a_t) = \sum_z p(a_{t+1}|z)p(z|a_t)$. Note that, to the extent that some function $g$ satisfies Assumption 1.1, we would also expect the value of $g$ to change slowly along trajectories of this chain, i.e. that $g(a_1) \approx g(a_2) \approx g(a_3) \approx \cdots$, and thus that in general $g(a_t) \approx \mathbb{E}_{p_+(a_{t+1}|a_t)}\big[g(a_{t+1})\big]$. This motivates solving for the *eigenfunctions* of the Markov chain, which are functions that satisfy $\mathbb{E}_{p_+(a_{t+1}|a_t)}\big[f_i(a_{t+1})\big] = \lambda_i f_i(a_t)$ for some $\lambda \in [0, 1]$.

As shown by Levin & Peres (2017, Chapter 12), the $f_i$ form an orthonormal basis[3] for the set of all functions $\mathcal{A} \to \mathbb{R}$ under the inner product $\langle f, g \rangle = \mathbb{E}_{p(a)}[f(a)g(a)]$, in the sense that $\mathbb{E}_{p(a)}[f_i(a)f_i(a)] = 1$ and $\mathbb{E}_{p(a)}[f_i(a)f_j(a)] = 0$ for $i \neq j$. Then, for any $g : \mathcal{A} \to \mathbb{R}$, if we let $c_i = \mathbb{E}_{p(a)}[f_i(a)g(a)]$, we must have $g(a) = \sum_i c_i f_i(a)$ and $\mathbb{E}_{p(a)}[g(a)^2] = \sum_i c_i^2$. Furthermore, this particular choice of orthonormal basis has the following appealing property:

**Proposition 3.1.** *If $g : \mathcal{A} \to \mathbb{R}$ and $c_i = \mathbb{E}[f_i(a)g(a)]$, then*

$$\mathbb{E}_{p_+(a_1, a_2)}\Big[\big(g(a_1) - g(a_2)\big)^2\Big] = \sum_i (2 - 2\lambda_i)c_i^2. \tag{3}$$

See Appendix C for a proof, along with a derivation of the orthonormality of the basis. One particular consequence of this fact is that the eigenfunctions with eigenvalues closest to 1 are also the most view-invariant. Specifically, setting $g = f_i$ reveals that

$$\mathbb{E}_{p_+(a_1, a_2)}\Big[\big(f_i(a_1) - f_i(a_2)\big)^2\Big] = 2 - 2\lambda_i. \tag{4}$$

More generally, if $g$ satisfies Assumption 1.1, Equation 3 implies that $2\sum_i(1 - \lambda_i)c_i^2 \leq \varepsilon$, and thus $g$ must have coefficients $c_i$ concentrated on eigenfunctions with $\lambda_i$ close to 1. Indeed, if $\varepsilon = 0$ (i.e. if $g$ is perfectly invariant to augmentations) then the only eigenfunctions with nonzero weights must be those with $\lambda_i = 1$. If we want to approximate $g$ using a small finite-dimensional representation, we should then prefer representations that allow us to estimate any linear combination of the eigenfunctions for which $\lambda_i$ is close to 1.

### 3.3 KERNEL PCA RECOVERS THE BASIS OF POSITIVE-PAIR EIGENFUNCTIONS

Surprisingly, it turns out that the representation built from kernel PCA in Section 3.1 has precisely the desired property. In fact, performing kernel PCA with $K_+$ (over the full population) is *exactly equivalent* to identifying the eigenfunctions of the Markov transition matrix $P$.

**Theorem 3.2.** *The output $(h_1, \sigma_1^2), (h_2, \sigma_2^2), \ldots$ of population-level Kernel PCA under $K_+$ and the orthonormal basis of eigenfunctions $f_i$ of $P$ with eigenvalues $\lambda_i$ satisfy $\sigma_i^2 = \lambda_i$ and $h_i(a) = \sigma_i f_i(a) = \lambda_i^{1/2} f_i(a)$ for all $i$ and all $a \in \mathcal{A}$ (up to reordering and multiplicity of eigenspaces[4]).*

See Appendix C for a proof. This theorem reveals a deep connection between the (co)variance of the dataset under our kernel $K_+$ and the view-invariance captured by Assumption 1.1. In particular, if we build a representation $r(a) = [h_1(a), h_2(a), \ldots, h_d(a)]$ using the first $d$ principal component projection functions, this representation will directly capture the $d$ eigenfunctions with eigenvalues closest to 1, and allow us to approximate any linear combination of those eigenfunctions using a linear predictor.

## 4 EIGENFUNCTION REPRESENTATIONS ARE MINIMAX OPTIMAL

We now give a more precise analysis of the quality of this representation for downstream supervised fine-tuning. We focus on the class of linear predictors on top of a $k$-dimensional representation

---

[3]As long as we scale them appropriately and choose orthogonal functions within each eigenspace.

[4]In other words, when some eigenvalues have multiplicity $> 1$, the $h_i$ and $f_i$ are not uniquely determined, but we are free to choose them such that they satisfy this relationship.

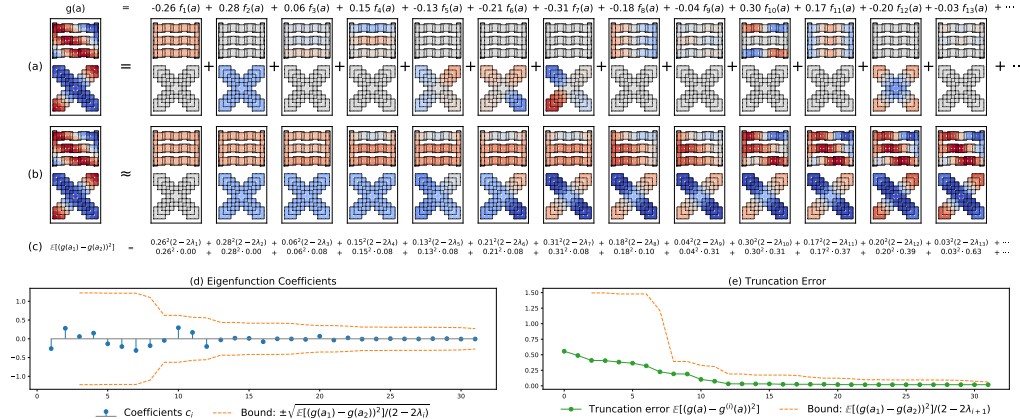

Figure 3: (a) We can expand an arbitrary function $g : \mathcal{A} \to \mathbb{R}$ as a linear combination of eigenfunctions, here using the toy problem described in Section 6 and Figure 4. (b) Taking partial sums using only the first $d$ eigenfunctions yields a series of increasingly good approximations of $g$. (c) The positive-pair discrepancy of $g$ is a linear combination of the eigenvalues of the corresponding eigenfunctions. (d) Under Assumption 1.1, we can bound each coefficient based on its contribution to the total positive-pair discrepancy. (e) This can be used to bound the error of approximating $g$ with a small subset of eigenfunctions. By Theorem 4.1, this ordered basis admits the smallest such bound.

$r : \mathcal{A} \to \mathbb{R}^d$, e.g. we will approximate $g$ with a parameterized function $\hat{g}_\beta(a) = \beta^\top r(a)$. It turns out that the representation consisting of the $d$ eigenfunctions $\{f_1, \ldots, f_d\}$ with largest eigenvalues $\{\lambda_1, \ldots, \lambda_d\}$ is the best choice under two simultaneous criteria.

**Theorem 4.1.** *Let $\mathcal{F}_r = \{a \mapsto \beta^\top r(a) : \beta \in \mathbb{R}^d\}$ be the subspace of linear predictors from representation $r$, and $S_\varepsilon$ be the set of functions satisfying Assumption 1.1. Let $r_*^d(a) = [f_1(a), f_2(a), \ldots, f_d(a)]$ be the representation consisting of the $d$ eigenfunctions of the positive pair Markov chain with the largest eigenvalues. Then $\mathcal{F}_{r_*^d}$ maximizes the view invariance of the least-invariant unit-norm predictor in $\mathcal{F}_{r_*^d}$:*

$$\mathcal{F}_{r_*^d} = \underset{\dim(\mathcal{F})=d}{\mathrm{argmin}} \; \underset{\hat{g} \in \mathcal{F}, \, \mathbb{E}[\hat{g}(a)^2]=1}{\max} \; \mathbb{E}_{p_+}\left[\left(\hat{g}(a_1) - \hat{g}(a_2)\right)^2\right]. \tag{5}$$

*Simultaneously, $\mathcal{F}_{r_*^d}$ minimizes the (quadratic) approximation error for the worst-case target function satisfying Assumption 1.1 for any fixed $\varepsilon$:*

$$\mathcal{F}_{r_*^d} = \underset{\dim(\mathcal{F})=d}{\mathrm{argmin}} \; \underset{g \in S_\varepsilon}{\max} \; \underset{\hat{g} \in \mathcal{F}}{\min} \; \mathbb{E}_{p(a)}\left[\left(g(a) - \hat{g}(a)\right)^2\right]. \tag{6}$$

Equation 5 states that the function class $\mathcal{F}_{r_*^d}$ has an implicit regularization effect: it contains the functions that change as little as possible over positive pairs, relative to their norm. Equation 6 reveals that this function class is also the optimal choice for least-squares approximation of a function satisfying Assumption 1.1. Together, these findings suggest that this representation should perform well as long as Assumption 1.1 holds.

We make this intuition precise as follows. Consider the loss function $\ell(\hat{y}, y) := |\hat{y} - y|$, and the associated risk $\mathcal{R}(g) = \mathbb{E}\left[\ell(g(A), Y)\right]$ of a predictor $g : \mathcal{A} \to \mathbb{R}$. Let $g^* \in \mathrm{argmin}\,\mathcal{R}(g)$ be the lowest-risk predictor over the set of functions $\mathcal{A} \to \mathbb{R}$, with risk $\mathcal{R}^* = \mathcal{R}(g^*)$, and assume it satisfies Assumption 1.1. Expand $g^*$ as a linear combination of the basis of eigenfunctions $(f_i)_{i=1}^{|\mathcal{A}|}$ as $g^* = \sum_{i=1}^{|\mathcal{A}|} \beta_i^* f_i$, and define the vector $\beta^* := (\beta_1^*, \ldots, \beta_d^*)$ by taking the first $d$ coefficients.

**Proposition 4.2.** *Let $(A_i, Y_i)_{i=1}^n$ be i.i.d. samples, choose $R \geq 0$, and consider the constrained empirical risk minimizer $\hat{\beta}_R \in \mathrm{argmin}_{\|\beta\|_2 \leq R} \, n^{-1} \sum_{i=1}^n |\langle \beta, r_d^*(A_i) \rangle - Y_i|$. Then the expected excess risk of $\hat{\beta}_R$ is bounded by:*

$$\mathbb{E}\left[\mathcal{E}(\hat{\beta}_R)\right] \leq \frac{2dR}{\sqrt{n}} + \sqrt{d}(\|\beta^*\|_2 - R)_+ + \sqrt{\frac{\varepsilon}{2(1 - \lambda_{d+1})}}$$

*where $\mathcal{E}(\beta) := \mathcal{R}(\beta) - \mathcal{R}^*$ is the excess risk and $(x)_+ := \max\{x, 0\}$.*

Note that $\|\beta^*\|_2^2 = \sum_{i=1}^d \beta_i^{*2} \leq \sum_{i=1}^{|\mathcal{A}|} \beta_i^{*2} = \mathbb{E}[g^*(a)^2]$, so if we choose $R^2 \geq \mathbb{E}[g^*(a)^2]$ then the second term vanishes. The third term bounds the error incurred by using only the first $d$ eigenfunctions, since $\beta_i^{*2} \leq \frac{\varepsilon}{2(1-\lambda_i)}$ by Equation 3, and motivates choosing $d$ to be large enough that $\lambda_{d+1}$ is small. See Appendix D for proofs of Theorem 4.1 and Proposition 4.2.

## 5 RELATED WORK

Our work is closely connected to the spectral graph theory analysis by HaoChen et al. (2021), which focuses on the eigenvectors of the normalized adjacency matrix of an augmentation graph, introduces the spectral contrastive loss, and shows that its optimum recovers the top eigenvectors up to an invertible transformation. We go further by arguing that the Spectral Contrastive, NT-XEnt, and NT-Logistic losses can all be viewed as approximating $K_+$, and showing that the resulting eigenfunctions are minimax-optimal for reconstructing target functions satisfying Assumption 1.1. (Indeed, the eigenvectors analyzed by HaoChen et al. are equal to our eigenfunctions scaled by $p(a)^{1/2}$; see Appendix C.3.) Our work is also related to the analysis of non-linear CCA given by Lee et al. (2020), which can be seen as an asymmetric variant of Kernel PCA with $K_+$. We note that Assumption 1.1 can be recast in terms of the Laplacian matrix of the augmentation graph; similar assumptions have been used before for label propagation (Bengio et al., 2006) and Laplacian filtering (Zhou & Srebro, 2011). This assumption is also used as a "consistency regularizer" for semi-supervised learning (Sajjadi et al., 2016; Laine & Aila, 2016) and self-supervised learning (Bardes et al., 2021).

There have been a number of other attempts to unify different contrastive learning techniques in a single theoretical framework. Balestriero & LeCun (2022) describe a unification using the framework of spectral embedding methods, and draw connections between SimCLR and Kernel ISOMAP. Tian (2022) provides a game-theoretic unification, and shows that contrastive losses are related to PCA in the *input* space for the special case of deep linear networks.

Techniques such as SpIN (Pfau et al., 2018) and NeuralEF (Deng et al., 2022) have been proposed to learn spectral decompositions of kernels. When applied to the positive-pair kernel $K_+$, it is possible to rewrite their objective in terms of paired views instead of requiring kernel evaluations, and the resulting decomposition is exactly the orthogonal basis of Markov chain eigenfunctions $f_i$. Interestingly, modifying Neural EF in this manner yields an algorithm that closely resembles the Variance-Invariance-Covariance regularization (VICReg) self-supervised learning method proposed by Bardes et al. (2021), as we discuss in Appendix E.2. See also Appendix B.4 for discussion of other connections between the positive-pair kernel and objectives considered by prior work.

## 6 EXPERIMENTS

It remains to determine how well learned approximations of the positive-pair kernel succeed at recovering the eigenfunction basis in practice. We explore this question on two synthetic testbed tasks for which the true kernel $K_+$ can be computed in closed form.

**Datasets.** Our first dataset is a simple "overlapping regions" toy problem, visualized in Figure 4. We define $\mathcal{A}$ to be a set of grid points, and $\mathcal{Z}$ to be a set of rectangular regions over the grid (shaded). We set $p(Z)$ to be a uniform distribution over regions, and $p(A|Z = z)$ to choose one grid point contained in $z$ at random. For a more natural distribution of data, our second dataset is derived from MNIST (LeCun et al., 2010), but with a carefully-chosen augmentation process so that computing $K_+$ is tractable. Specifically, we choose $p(Z)$ to uniformly select from a small subset $\mathcal{Z}$ of MNIST digits, and define $p(A|Z = z)$ by first transforming $z$ using one of a finite set of possible rotations and translations, then sampling a subset of $k$ pixels with replacement from the transformed copy. The finite set of allowed transformations and the tractable probability mass function of the multinomial distribution together enable us to compute $K_+$ by summing over all possible $z \in \mathcal{Z}$. (Samples from this distribution for different values of $k$ are shown in Figure 2.)

**Model training and eigenfunction estimation.** We train contrastive learning models for each of these datasets using a variety of kernel parameterizations and loss functions. For kernel parameterizations, we consider both **linear** approximate kernels $\widehat{K}_\theta(a_1, a_2) = h_\theta(a_1)^\top h_\theta(a_2)$, where $h_\theta$ may be normalized to have constant norm $\|h_\theta(a)\|^2 = c$ or be left unconstrained, and **hypersphere-**

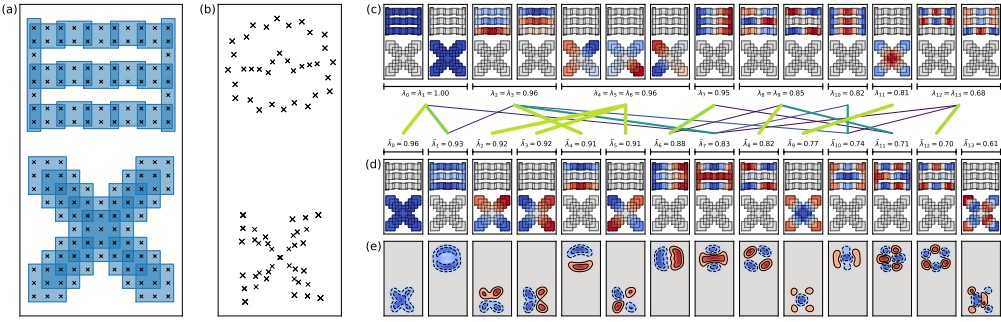

Figure 4: **(a)** Toy "overlapping regions" contrastive learning task, where $\mathcal{A}$ is the set of cross markers, and $\mathcal{Z}$ is the set of shaded blue rectangles. **(b)** 2D embedding space learned by minimizing a contrastive loss with a rational quadratic kernel head $\widehat{K}_\theta$. **(c)** The first 14 eigenfunctions of the true positive-pair Markov chain. **(d)** The first 14 principal component projection functions for the learned kernel $\widehat{K}_\theta$ extracted using Kernel PCA. Diagonal colored lines denote alignment between the learned functions and the true eigenspaces, which increases as $\widehat{K}$ approaches $K_+$. **(e)** Evaluations of the the learned projection functions over the entire latent embedding space using Kernel PCA, showing that it can embed points not seen during training.

based approximate kernels $\widehat{K}_\theta(a_1, a_2) = \exp(h_\theta(a_1)^\top h_\theta(a_2)/\tau + b)$, where $\|h_\theta(a)\|^2 = 1$ and $b \in \mathbb{R}, \tau \in \mathbb{R}^+$ are learned. We also explore either replacing $b$ with a learned per-example adjustment $s_\theta(a_1) + s_\theta(a_2)$ or fixing it to zero. For losses, we investigate the XEnt, Logistic, and Spectral losses shown in Table 1, and explore using a downweighted Logistic loss as a regularizer for the XEnt loss to eliminate the underspecified proportionality constant in minimizer of the XEnt loss.

For each approximate kernel and also for the true positive-pair kernel $K_+$, we apply Kernel PCA to extract the eigenfunctions $\hat{f}_i$ and their associated eigenvalues $\hat{\lambda}_i$. We use full-population Kernel PCA for the overlapping regions task, and combine the Nyström method (Williams & Seeger, 2000) with a large random subset of augmented examples to approximate Kernel PCA for the MNIST task. We additionally investigate training a Neural EF model (Deng et al., 2022) to directly decompose the positive pair kernel into principal components using a single model instead of separately training a contrastive learning model, as mentioned in Section 5. We modify the Neural EF objective slightly to make it use positive pair samples instead of kernel evaluations and to increase numerical stability. See Appendix E for additional discussion of the eigenfunction approximation process.

We then measure the alignment of each eigenfunction $\hat{f}_i$ of our learned models with the corresponding eigenfunction $f_j$ of $K_+$ using the formula $\mathbb{E}_{p(a)}[\hat{f}_i(a)f_j(a)]^2$, where the square is taken since eigenfunctions are invariant to changes in sign. We also estimate the positive-pair discrepancy $\mathbb{E}_{p_+}\left[(\hat{f}_i(a_1) - \hat{f}_i(a_2))^2\right]$ for each approximate eigenfunction.

**Results.** We summarize a set of qualitative observations which are relevant to our theoretical claims in the previous sections. A representative subset of the results are also visualized in Figure 5. See Appendix F.1 for additional experimental results and a more thorough discussion of our findings.

*Linear kernels, hypersphere kernels, and NeuralEF can all produce good approximations of the basis of eigenfunctions* with sufficient tuning, despite their different parameterizations. The relationship between approximate eigenvalues and positive-pair discrepancies also closely matches the prediction from Equation 4. Both of these relationships emerge during training and do not hold for a randomly initialized model. Additionally, for a fixed kernel parameterization, *multiple losses can work well*. In particular, we were able to train good hypersphere-based models on the toy regions task using either the XEnt loss or the spectral loss, although the latter required additional tuning to stabilize learning.

*Constraints on the kernel approximation degrade eigenfunction and eigenvalue estimates.* We find that introducing constraints on the output head tends to produce worse alignment between eigenfunctions, and leads to eigenvalues that deviate from the expected relationship in Equation 4. Such constraints include reducing the dimension of the output layer, rescaling the output layer for a linear kernel parameterization to have a fixed L2 norm (as proposed by HaoChen et al. (2021)), or fixing $b$ to zero for a hypersphere kernel parameterization (as is done implicitly by Chen et al. (2020a)).

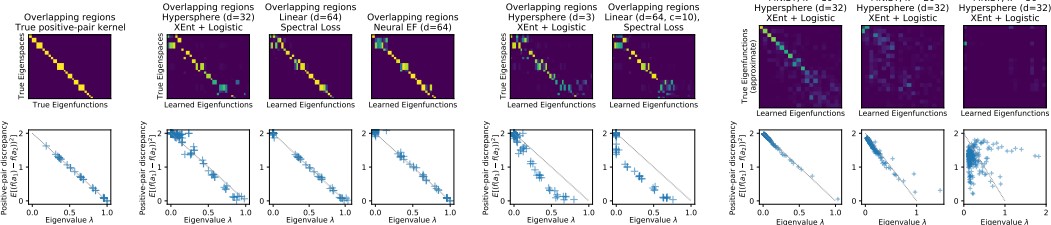

Figure 5: Eigenfunction and eigenvalue estimation accuracy for a selection of models on our two synthetic tasks. Top row: Alignment between the eigenfunctions of $K_+$ and those of each kernel approximation, with perfect alignment shown as a block diagonal matrix. Bottom row: Relationship between learned kernel eigenvalue $\lambda$ and the corresponding positive-pair discrepancy $\mathbb{E}_{p_+}\left[(f(a_1) - f(a_2))^2\right]$, with the relationship predicted by Equation 4 shown with a dashed line.

*Weaker augmentations make eigenfunction estimation more difficult.* Finally, for the MNIST task, we find that as we decrease the augmentation strength (by increasing the number of kept pixels $k$), the number of successfully-recovered eigenfunctions also decreases. This may be partially due to Kernel PCA having worse statistical behavior when there are smaller gaps between eigenvalues (specifically, more eigenvalues close to 1), since we observe this trend even when applying Kernel PCA to the exact closed-form $K_+$. However, we also observe that the relationship predicted by Equation 4 starts to break down as well for some kernel approximations under weak augmentation strengths.

## 7 DISCUSSION

We have shown that existing contrastive objectives approximate the kernel $K_+$, and that Kernel PCA yields an optimal representation for linear prediction under Assumption 1.1. We have further demonstrated on two synthetic tasks that running contrastive learning followed by Kernel PCA can yield good approximate representations across multiple parameterizations and losses, although constrained parameterizations and weaker augmentations both reduce approximation quality.

Our analysis (in particular Theorem 4.1) assumes that the distribution of views $p(A)$ is shared between the contrastive learning task and the downstream learning task. In practice, the distribution of underlying examples $p(Z)$ and the augmentation distribution $p(A|Z)$ often change when fine-tuning a self-supervised pretrained model. An interesting future research direction would be to quantify how the minimax optimality of our representation is affected under such a distribution shift. Our analysis also focuses on the standard "linear evaluation protocol", which determines representation quality based on the accuracy of a linear predictor. This measurement of quality may not be directly applicable to tasks other than classification and regression (e.g. object detection and segmentation), or to other downstream learning methods (e.g. fine-tuning or $k$-nearest-neighbors). In these other settings, our theoretical framework is not directly applicable, but we might still hope that the view-invariant features arising from Kernel PCA with $K_+$ would be useful.

Additionally, our analysis precisely characterizes the optimal representation if all we know about our target function $g$ is that it satisfies Assumption 1.1, but in practice we often have additional knowledge about $g$. In particular, Saunshi et al. (2022) show that *inductive biases* are crucial for building good representations, and argue that standard distributions of augmentations are approximately disjoint, producing many eigenvalues very close to one. Interestingly, this is exactly the regime where the correspondence between the approximate kernels and the positive-pair kernel $K_+$ begins to break down in our experiments. We believe this opens up a number of exciting opportunities for research, including studying the training dynamics of parameterized kernels $\widehat{K}_\theta$ under a weak augmentation regime, analyzing the impact of additional assumptions about $g$ and inductive biases in $\widehat{K}_\theta$ on the minimax optimality of PCA representations, and exploring new model parameterizations and objectives that trade off between inductive biases and faithful approximation of $K_+$.

More generally, the authors believe that the connections drawn in this work between contrastive learning, kernel methods, principal components analysis, and Markov chains provide a useful lens for theoretical study of self-supervised representation learning and also give new insights toward building useful representations in practice.

ACKNOWLEDGEMENTS

We would like to thank David Duvenaud, Daniel Tarlow, and Lechao Xiao for providing feedback on this manuscript, and the ICLR 2023 reviewers for their additional suggestions and comments. We would also like to thank the reviewers and organizers of the ICML 2022 workshop on Pre-training: Perspectives, Pitfalls, and Paths Forward for providing a venue for an earlier version of this work.

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

APPENDIX

# A  JUSTIFICATION OF ASSUMPTION 1.1

In this section we expand on the justification of Assumption 1.1 given in Section 1.

Recall that during contrastive learning we have a distribution $p(Z)$ of original examples, and a distribution $p(A|Z = z)$ of augmented views for each original example $z$. Then, at downstream supervised learning time, we additionally have a distribution $p(Y = y|Z = z)$ of labels $y \in \mathbb{R}$ for each original example $z$. (For simplicity we assume that the distribution of $p(Z)$ remains unchanged during the downstream learning step. We also assume $y$ is a scalar; the vector case can be derived by independently estimating each element of $Y$ with a separate target function.)

Since the distribution of augmented views is typically defined via a random perturbation of $Z$ without using $Y$, the augmented views $A$ are conditionally independently of the label $Y$ given the original example $Z$, so we have joint distribution $p(a, y, z) = p(z)p(a|z)p(y|z)$. When we draw a pair of augmented views for the same original example, we instead have a joint $p(a_1, a_2, y, z) = p(z)p(a_1|z)p(a_2|z)p(y|z)$. We also have some target function $g : \mathcal{A} \to \mathbb{R}$ we are trying to approximate. For instance, we might want to approximate a Bayes-optimal predictor of $Y$ under some loss function, such as $g(a) = \mathbb{E}[Y|A = a]$ for the quadratic loss.

Common wisdom states that $p(A|Z)$ should remove irrelevant information from $Z$ without removing (much) information about $Y$ (most of the time). This means that, if augmentations are chosen appropriately, we can expect $\mathbb{E}[(g(A) - Y)^2]$ to be small under our joint probability distribution for an appropriate choice of $g$.

We can then expand the left-hand side of Assumption 1.1 as

$$\mathbb{E}[(g(A_1) - g(A_2))^2] = \mathbb{E}[((g(A_1) - Y) - (g(A_2) - Y))^2]$$
$$= \mathbb{E}[(g(A_1) - Y)^2 - 2(g(A_1) - Y)(g(A_2) - Y) + (g(A_2) - Y)^2]$$
$$= 2\mathbb{E}[(g(A) - Y)^2] - 2\mathbb{E}[(g(A_1) - Y)(g(A_2) - Y)]$$

Furthermore, due to the law of total expectation combined with our conditional independence assumptions, we know that

$$\mathbb{E}[(g(A_1) - Y)(g(A_2) - Y)] = \mathbb{E}_{Z,Y}[\mathbb{E}_{A_1,A_2}[(g(A_1) - Y)(g(A_2) - Y)|Z, Y]]$$
$$= \mathbb{E}_{Z,Y}[\mathbb{E}_{A_1,A_2}[(g(A_1) - Y)|Z, Y]\mathbb{E}_{A_1,A_2}[(g(A_2) - Y)|Z, Y]]$$
$$= \mathbb{E}_{Z,Y}[\mathbb{E}_A[(g(A) - Y)|Z, Y]^2] \geq 0.$$

We can then conclude that

$$\mathbb{E}[(g(A_1) - g(A_2))^2] = 2\mathbb{E}[(g(A) - Y)^2] - 2\mathbb{E}[(g(A_1) - Y)(g(A_2) - Y)]$$
$$\leq 2\mathbb{E}[(g(A) - Y)^2]$$

Thus, Assumption 1.1 holds whenever $\mathbb{E}[(g(A) - Y)^2] \leq \frac{1}{2}\varepsilon$. Intuitively, if it is possible to estimate $Y$ with high accuracy using a deterministic function, then that function must satisfy Assumption 1.1.

Note that Assumption 1.1 itself may still hold even if $g$ is not a good estimate of $Y$, and is well defined even if we never specify $Y$ and work only with the joint distribution $p(A_1, A_2)$. This is particularly useful because it allows us to quantify over all possible choices of $g$ in a sensible way with minimal knowledge about the label distribution itself.

## B EXISTING OBJECTIVES ARE MINIMIZED BY THE POSITIVE-PAIR KERNEL

In this section, we show that the positive-pair kernel is the optimum of the objectives shown in Table 1 (although this optimum is not unique for cross-entropy loss), and discuss some connections to other objectives considered by previous work. Throughout this section, we use $p(Z)$ to denote the true data distribution over unperturbed examples, $p(A|Z)$ to denote the distribution of augmented views conditioned on a particular unperturbed example, and $p(A)$ to denote the marginal distribution of augmented views, e.g.

$$p(A = a) = \sum_{z \in \mathcal{Z}} p(Z = z)p(A = a|Z = z).$$

We use $p_+(A_1, A_2)$ to denote the positive pair distribution induced by $p(Z)$ and $p(A|Z)$, defined by

$$p_+(A_1 = a_1, A_2 = a_2) = \sum_{z \in \mathcal{Z}} p(Z = z)p(A = a_1|Z = z)p(A = a_2|Z = z).$$

For notational convenience, we will use shorthand $p(z)$ for $p(Z = z)$, $p(a)$ for $p(A = a)$, $p_+(a_1, a_2)$ for $p_+(A_1 = a_1, A_2 = a_2)$, and so on.

### B.1 NT-XENT AND THE INFONCE OBJECTIVE

The NT-XEnt objective described by Chen et al. (2020a) (and previously used by Sohn (2016); Van den Oord et al. (2018); Wu et al. (2018)) has the form

$$\mathcal{L}_{\text{NT-Xent}}(\theta) = \mathbb{E}_{\substack{(a_1^+, a_2^+) \sim p_+(A_1, A_2) \\ a_i^- \sim p(A)}} \left[ -\log \frac{\exp\left(\frac{h_\theta(a_1^+)^\top h_\theta(a_2^+)}{\tau}\right)}{\exp\left(\frac{h_\theta(a_1^+)^\top h_\theta(a_2^+)}{\tau}\right) + \sum_{a_i^-} \exp\left(\frac{h_\theta(a_1^+)^\top h_\theta(a_i^-)}{\tau}\right)} \right].$$

For simplicity, we assume that all of the negative samples $a_i^-$ are drawn independently from the marginal distribution when computing the loss for $a_1^+$ and $a_2^+$. (In practice, implementations often generate negative samples by taking elements of other positive pairs, e.g. $(a_1^-, a_2^-) \sim p_+(a_1^-, a_2^-)$, $(a_3^-, a_4^-) \sim p_+(a_3^-, a_4^-)$, and so on.)

We can decompose this objective into two parts: an InfoNCE-like loss (Van den Oord et al., 2018)

$$\mathcal{L}_{\text{InfoNCE}}(\widehat{K}_\theta) = \mathbb{E}_{(a_1^+, a_2^+) \sim p_+(A_1, A_2), a_i^- \sim p(A)} \left[ -\log \frac{\widehat{K}_\theta(a_1^+, a_2^+)}{\widehat{K}_\theta(a_1^+, a_2^+) + \sum_{a_i^-} \widehat{K}_\theta(a_1^+, a_i^-)} \right]$$

combined with a particular parameterized function

$$\widehat{K}_\theta(a_1, a_2) = \exp\left(\frac{h_\theta(a_1)^\top h_\theta(a_2)}{\tau}\right).$$

where $h_\theta : \mathcal{A} \to \mathbb{R}^{n+1}$ maps inputs to points on the $n$-dimensional hypersphere.

We first observe that $\widehat{K}_\theta(a_1, a_2)$ defines a positive definite kernel, within the family of "dot product kernels". Indeed, when $h_\theta$ is restricted to the unit hypersphere, this parameterization is equivalent to the squared-exponential kernel (also called radial-basis-function kernel)

$$\widehat{K}_\theta(a_1, a_2) = \exp\left(\frac{1 - \frac{1}{2}\|h_\theta(a_1) - h_\theta(a_2)\|^2}{\tau}\right) = \exp(1/\tau) \exp\left(-\frac{\|h_\theta(a_1) - h_\theta(a_2)\|^2}{2\tau}\right)$$

using the identity $\|h_\theta(a_1) - h_\theta(a_2)\|^2 = 2 - 2h_\theta(a_1)^\top h_\theta(a_2)$. Although this kernel is positive definite, it has "infinite dimension" and cannot be expressed as an inner product of finite-dimensional embedding vectors; nevertheless, we can still run algorithms such as Kernel PCA on a finite dataset. (See Bach (2021, Chapter 7) for some additional background on positive-definite kernels, and Scetbon & Harchaoui (2021) for discussion of other dot product kernels on the unit hypersphere.)

We next discuss the minimum of the NT-XEnt objective, under the unconstrained setting where we allow $\widehat{K}_\theta(a_1, a_2)$ to be an arbitrary symmetric function. The InfoNCE loss, in its more general form,

is not necessarily symmetric: it is based on a distribution of contexts $p(C)$, positive samples $p(A|C)$, and negative samples drawn from the marginal distribution $P(A)$, and is given by

$$\mathcal{L}_{\text{InfoNCE}}(f_\theta) = \mathbb{E}_{c \sim p(c), a^+ \sim p(A|C=c), a_i^- \sim p(A)} \left[ -\log \frac{f_\theta(c, a^+)}{f(c, a^+) + \sum_{a_i^-} f_\theta(c, a_i^-)} \right]$$

Van den Oord et al. (2018) show that every minimizer of this objective is of the form

$$f_*(c, a) = \frac{p(a|c)}{p(a)} \cdot b(c) = \frac{p(a, c)}{p(a)p(c)} \cdot b(c)$$

for some function $b(c)$ that does not depend on $a$. In other words, holding $c$ fixed, $f_*(c, a) \propto \frac{p(a,c)}{p(a)p(c)}$. Intuitively, this is because the exact probability of $(c, a^+)$ being the positive pair given $c$ and the set $\{a^+, a_1^-, \ldots, a_K^-\}$ is also proportional to this density ratio, and the InfoNCE objective is a cross-entropy objective for identifying the positive pair. (See also Poole et al. (2019) for a different proof.)

In the case of the NT-XEnt contrastive objective, we choose the context $C$ to be one of the augmentations $A_1^+$, and the positive sample to be the other augmentation $A_2^+$ drawn according to $p_+(A_2^+|A_1^+)$. We furthermore restrict our attention to symmetric functions $\widehat{K}_\theta$, e.g. functions for which $\widehat{K}_\theta(a_1, a_2) = \widehat{K}_\theta(a_2, a_1)$. In this case, the minimizer is

$$\widehat{K}_*(a_1, a_2) = \frac{p_+(a_1, a_2)}{p(a_1)p(a_2)} \cdot b(a_1) = \frac{p_+(a_1, a_2)}{p(a_1)p(a_2)} \cdot b(a_2).$$

so we must have $b(a_1) = b(a_2)$ for any pair $(a_1, a_2)$ for which $p_+(a_1, a_2) > 0$.

If we assume that the positive pair Markov chain is irreducible, e.g. that there is a single communicating class and it is possible to reach any augmentation in $\mathcal{A}$ from any other augmentation over a long enough trajectory, then the function $b(a)$ must be constant everywhere, and thus

$$\widehat{K}_*(a_1, a_2) = f_*(a_1, a_2) = \frac{p_+(a_1, a_2)}{p(a_1)p(a_2)} \cdot B$$

for some $B \in \mathbb{R}^+$. In this case, $\widehat{K}_*$ is equivalent to $K_+$ up to a scaling constant.

If the Markov chain has multiple communicating classes (e.g. if the augmentations can be partitioned so that augmentations always come from the same partition), the function $b(a)$ may assign a different value to different communicating classes. Nevertheless, any such minimizer is still a kernel, since we can write it as

$$\widehat{K}_*(a_1, a_2) = \left\langle \sqrt{b(a_1)} \begin{bmatrix} \frac{p(a_1|z_1)\sqrt{p(z_1)}}{p(a_1)} \\ \frac{p(a_1|z_2)\sqrt{p(z_2)}}{p(a_1)} \\ \vdots \\ \frac{p(a_1|z_{|\mathcal{Z}|})\sqrt{p(z_{|\mathcal{Z}|})}}{p(a_1)} \end{bmatrix}, \ \sqrt{b(a_2)} \begin{bmatrix} \frac{p(a_2|z_1)\sqrt{p(z_1)}}{p(a_2)} \\ \frac{p(a_2|z_2)\sqrt{p(z_2)}}{p(a_2)} \\ \vdots \\ \frac{p(a_2|z_{|\mathcal{Z}|})\sqrt{p(z_{|\mathcal{Z}|})}}{p(a_2)} \end{bmatrix} \right\rangle$$

$$= \left\langle \sqrt{b(a_1)} \cdot \phi_+(a_1), \ \sqrt{b(a_2)} \cdot \phi_+(a_2) \right\rangle$$

Indeed, such a minimizer is equivalent to $K_+$ except that it scales the inner product by the value of $b(a)$ for each communicating class. It is still possible to extract the set of Markov chain eigenfunctions from the set of principal components of this kernel, although one must correct for the scaling factor when computing the eigenvalues; see Appendix C.4 for details. Alternatively, one can ensure a unique minimum by combining the NT-Xent/InfoNCE loss with either the spectral or logistic losses (discussed below).

## B.2 Logistic losses and NT-Logistic

Logistic losses have also been proposed for contrastive learning, including the NT-Logistic objective as described in Chen et al. (2020a) and other versions described by Mikolov et al. (2013) and Tosh

et al. (2021). Such losses take the form

$$\mathcal{L}_{\text{Logistic}}(f_\theta) = \mathbb{E}_{(a_1^+, a_2^+) \sim p_+(A_1, A_2)} \left[ -\log \sigma(f_\theta(a_1^+, a_2^+)) \right]$$
$$+ \mathbb{E}_{a_1^- \sim p(A), a_2^- \sim p(A)} \left[ -\log \sigma(-f_\theta(a_1^-, a_2^-)) \right]$$

where negative samples are drawn independently from the marginal distribution $p(A)$. Tosh et al. (2021) motivates this loss based on a binary classification: choose a label $Y$ to be 0 or 1 with probability $1/2$ each, sample a positive pair $(a_1, a_2) \sim p_+(A_1, A_2)$ if $Y = 1$ and a negative pair $a_1 \sim p(A), a_2 \sim p(A)$ if $Y = 0$, then use a learned model to predict $Y$ given the pair. The minimizer of this loss is then the conditional log-odds-ratio

$$f_*(a_1, a_2) = \log \frac{p(Y = 1 | a_1, a_2)}{p(Y = 0 | a_1, a_2)} = \log \frac{p(a_1, a_2 | Y = 1) p(Y = 1)}{p(a_1, a_2 | Y = 0) p(Y = 0)}$$
$$= \log \frac{p_+(a_1, a_2) \cdot \frac{1}{2}}{p(a_1) p(a_2) \cdot \frac{1}{2}} = \log \frac{p_+(a_1, a_2)}{p(a_1) p(a_2)}.$$

For the particular case of the NT-Logistic objective, we parameterize $f_\theta$ as

$$f_\theta(a_1, a_2) = \log \hat{K}_\theta(a_1, a_2) = \frac{h_\theta(a_1)^\top h_\theta(a_2)}{\tau}$$

where we again define

$$\widehat{K}_\theta(a_1, a_2) = \exp\left( \frac{h_\theta(a_1)^\top h_\theta(a_2)}{\tau} \right).$$

The optimum (if we ignore the constraints of this particular form of $\hat{K}$ and minimize over all functions of two variables) is then

$$\hat{K}_*(a_1, a_2) = \frac{p_+(a_1, a_2)}{p(a_1) p(a_2)}.$$

Note that in this case there is no proportionality constant.

## B.3  THE SPECTRAL CONTRASTIVE LOSS

HaoChen et al. (2021) propose the Spectral Contrastive Loss as an alternative to other contrastive losses with provable performance guarantees. The loss is defined as

$$\mathcal{L}_{\text{Spectral}}(\widehat{K}_\theta) = -2 \cdot \mathbb{E}_{(a_1^+, a_2^+) \sim p_+(A_1, A_2)} \left[ \widehat{K}_\theta(a_1^+, a_2^+) \right] + \mathbb{E}_{a_1^- \sim p(A), a_2^- \sim p(A)} \left[ \widehat{K}_\theta(a_1^-, a_2^-)^2 \right]$$

where they choose

$$\widehat{K}_\theta(a_1, a_2) = h_\theta(a_1)^\top h_\theta(a_2).$$

for a learned embedding function $h_\theta : \mathcal{A} \to \mathbb{R}^d$. We note that this directly satisfies the definition of a kernel, in that it is an inner product in a transformed space. One interesting property of this kernel approximation is that it can be negative, whereas the exponential-based kernel approximations in the previous sections are always nonnegative.

HaoChen et al. show that this loss can be rewritten as

$$\mathcal{L}_{\text{Spectral}}(\widehat{K}_\theta) = \sum_{a_1, a_2} \left( -2 p_+(a_1, a_2) \widehat{K}_\theta(a_1, a_2) + p(a_1) p(a_2) \widehat{K}_\theta(a_1, a_2)^2 \right)$$
$$= \sum_{a_1, a_2} \left( \frac{p_+(a_1, a_2)^2}{p(a_1) p(a_2)} - 2 p_+(a_1, a_2) \widehat{K}_\theta(a_1, a_2) + p(a_1) p(a_2) \widehat{K}_\theta(a_1, a_2)^2 \right)$$
$$- \sum_{a_1, a_2} \frac{p_+(a_1, a_2)^2}{p(a_1) p(a_2)}$$
$$= \sum_{a_1, a_2} \left( \frac{p_+(a_1, a_2)}{\sqrt{p(a_1) p(a_2)}} - \sqrt{p(a_1) p(a_2)} \widehat{K}_\theta(a_1, a_2) \right)^2 - C,$$

where $C = \sum_{a_1, a_2} \frac{p_+(a_1, a_2)^2}{p(a_1) p(a_2)}$ is a constant independent of the model.

If we again ignore the constraints on $\hat{K}_\theta$, the minimum of the spectral loss must occur when

$$\frac{p_+(a_1, a_2)}{\sqrt{p(a_1)p(a_2)}} = \sqrt{p(a_1)p(a_2)}\widehat{K}_\theta(a_1, a_2),$$

for all $a_1$ and $a_2$, or in other words, when

$$\widehat{K}_\theta(a_1, a_2) = \frac{p_+(a_1, a_2)}{p(a_1)p(a_2)} = K_+(a_1, a_2).$$

HaoChen et al. continue by expanding their definition of $\widehat{K}_\theta$ for a fixed representation $d$, and showing that it relates to the spectral decomposition of a particular augmentation graph. It turns out that this decomposition is equivalent to our decomposition in terms of eigenfunctions except for a scaling factor of $p(a)^{1/2}$; we discuss this connection more in Appendix C.3.

### B.4 OTHER RELATED OBJECTIVES AND CONNECTIONS TO THE POSITIVE-PAIR KERNEL

We note that the log-probability ratio $\log \frac{p(u,v)}{p(u)p(v)}$ has an information-theoretic interpretation as the pointwise mutual information between two random variates $u$ and $v$. We can thus view the positive-pair kernel $K_+$ as being an exponentiated version of the pointwise mutual information between two views $A_1$ and $A_2$. This ratio has been shown to be an optimal critic for mutual information estimation (Poole et al., 2019; Nowozin et al., 2016; Hjelm et al., 2018).

Moustakides & Basioti (2019) describe several sample-based estimators for probability ratios between arbitrary densities or mass functions. These estimators can be seen as generalizations of the the contrastive losses in Table 1, with the goal of estimating the ratio between the positive and negative pair distributions.

The VICReg semi-supervised learning technique can be reinterpreted in terms of the positive-pair kernel as a particular form of kernel decomposition method. See Appendix E.2 for discussion of this connection.

Although the parameterized kernels in Table 1 have a fairly simple form, some prior work has considered more sophisticated parameterizations for learning kernels with neural networks (Wilson et al., 2016; Sun et al., 2018). There have also been recent works related to reinterpreting existing neural network models as kernels (Jacot et al., 2018; Shankar et al., 2020; Amid et al., 2022). It would be interesting to compare the properties of these other kernel parameterizations with the implicit kernels involved in contrastive learning methods.

Finally, we note that early approaches to contrastive learning such as DrLIM (Hadsell et al., 2006) were motivated in part by removing limitations of previous spectral embedding techniques, which required explicitly selecting an input-space distance metric or kernel function (e.g. Bengio et al. (2003)). Our analysis reveals that that, for modern contrastive learning methods, choosing the distribution of augmentations can still be seen as *implicitly* defining a kernel function of this form. Conveniently, however, we do not need to be able to evaluate this kernel to train contrastive learning models; we only need to sample augmentations.

## C  RELATIONSHIP BETWEEN POSITIVE-PAIR KERNEL AND MARKOV CHAIN EIGENFUNCTIONS

In this section, we describe the relationship between the positive-pair kernel principal components and the Markov chain eigenfunctions in more detail.

### C.1  NOTATION

We start by introducing some notation that will be useful.

Throughout, we will identify functions $f : \mathcal{A} \to \mathbb{R}$ as vectors $\mathbf{f} : \mathbb{R}^{\mathcal{A}}$, which will allow us to use matrix notation for many of the relevant quantities. We will also use $\mathbf{e}_i$ to represent the vector that has a one at the $i$th position and zeros in all other positions.

We will let

$$D_Z = \mathrm{diag}(p(z_1), p(z_2), \dots, p(z_{|\mathcal{Z}|})),$$
$$D_A = \mathrm{diag}(p(a_1), p(a_2), \dots, p(a_{|\mathcal{A}|})),$$

be diagonal matrices containing the marginal probabilities of each element in $\mathcal{Z}$ and $\mathcal{A}$, respectively, under the true data distribution. We will assume that the distribution has full support, and thus both $D_Z$ and $D_A$ are invertible. We also define the matrices

$$[P_{Z,A}]_{z,a} = p(z,a) \qquad [P_{Z \to A}]_{z,a} = p(a|z) \qquad [P_{Z \leftarrow A}]_{z,a} = p(z|a)$$
$$[P_{A,Z}]_{a,z} = p(z,a) \qquad [P_{A \leftarrow Z}]_{a,z} = p(a|z) \qquad [P_{A \to Z}]_{a,z} = p(z|a)$$

Equivalently

$$P_{Z \to A} = D_Z^{-1} P_{Z,A}, \qquad\qquad P_{Z \leftarrow A} = P_{Z,A} D_A^{-1},$$
$$P_{A \leftarrow Z} = (P_{Z \to A})^{\top}, \qquad\qquad P_{A \to Z} = (P_{Z \leftarrow A})^{\top}.$$

From these, we can construct the positive pair probability matrix

$$P_{A,A} = P_{A \leftarrow Z} \, D_Z \, P_{Z \to A}$$
$$[P_{A,A}]_{i,j} = p(A_1 = i, A_2 = j) = \sum_z p(A = i|z) p(z) p(A = j|z).$$

### C.2  PROOF OF CORRESPONDENCE BETWEEN KERNEL PCA AND MARKOV CHAIN EIGENFUNCTIONS

**The positive-pair kernel.**   Writing the positive-pair kernel $K_+$ in matrix form, such that $[K_+]_{i,j} = K_+(i,j)$, our definition $K_+(a_1, a_2) = \frac{p_+(a_1,a_2)}{p(a_1)p(a_2)}$ becomes the matrix equation

$$K_+ = D_A^{-1} P_{A,A} D_A^{-1}.$$

One way to expand $K_+$ is as a product

$$K_+ = D_A^{-1} P_{A \leftarrow Z} D_Z P_{Z \to A} D_A^{-1} = \left( D_Z^{1/2} P_{Z \to A} D_A^{-1} \right)^{\top} \left( D_Z^{1/2} P_{Z \to A} D_A^{-1} \right) = \Phi_+^{\top} \Phi_+$$

where $\Phi_+ \in \mathbb{R}^{\mathcal{Z} \times \mathcal{A}}$ is a matrix whose columns are given by $\phi_+$:

$$\Phi_+ = D_Z^{1/2} P_{Z \to A} D_A^{-1} = \begin{bmatrix} \phi_+(a_1) & \phi_+(a_2) & \dots & \phi_+(a_{|\mathcal{A}|}) \end{bmatrix}$$

$$= \begin{bmatrix} \frac{p(a_1|z_1)\sqrt{p(z_1)}}{p(a_1)} & \frac{p(a_2|z_1)\sqrt{p(z_1)}}{p(a_2)} & \dots & \frac{p(a_{|\mathcal{A}|}|z_1)\sqrt{p(z_1)}}{p(a_{|\mathcal{A}|})} \\[6pt] \frac{p(a_1|z_2)\sqrt{p(z_2)}}{p(a_1)} & \frac{p(a_2|z_2)\sqrt{p(z_2)}}{p(a_2)} & \dots & \frac{p(a_{|\mathcal{A}|}|z_2)\sqrt{p(z_2)}}{p(a_{|\mathcal{A}|})} \\[6pt] \vdots & \vdots & \ddots & \vdots \\[6pt] \frac{p(a_1|z_{|\mathcal{Z}|})\sqrt{p(z_{|\mathcal{Z}|})}}{p(a_1)} & \frac{p(a_2|z_{|\mathcal{Z}|})\sqrt{p(z_{|\mathcal{Z}|})}}{p(a_2)} & \dots & \frac{p(a_{|\mathcal{A}|}|z_{|\mathcal{Z}|})\sqrt{p(z_{|\mathcal{Z}|})}}{p(a_{|\mathcal{A}|})} \end{bmatrix}.$$

Equivalently, we have $\phi_+(a) = \Phi_+ \mathbf{e}_a$.

Since $K_+(a_1, a_2) = \phi_+(a_1)^\top \phi_+(a_2)$, $\phi_+$ is called a *feature map* for $K_+$. Note that there are multiple possible feature maps for $K_+$: given any orthonormal matrix $Q$, the function $\phi_Q(a) = Q\phi_+(a)$ is also a feature map for the kernel, since

$$\phi_Q(a_1)^\top \phi_Q(a_1) = \phi_+(a)^\top Q^\top Q\phi_+(a) = \phi_+(a)^\top \phi_+(a) = K_+(a_1, a_2).$$

Performing kernel PCA under $K_+$ is equivalent to performing ordinary PCA over any of its feature maps (since the principal component projection functions are independent of the particular feature map chosen). We thus focus on analyzing the principal component projection functions for the feature map $\phi_+$.

The population level principal components are the eigenvectors of the (uncentered) covariance matrix

$$\Sigma = \mathbb{E}_{p(a)}[\phi_+(a)\phi_+(a)^\top] = \mathbb{E}_{p(a)}[\Phi_+ \mathbf{e}_a \mathbf{e}_a^\top \Phi_+^\top] = \Phi_+ \mathbb{E}_{p(a)}[\mathbf{e}_a \mathbf{e}_a^\top]\Phi_+^\top = \Phi_+ D_A \Phi_+^\top.$$

Note that we are working with the *uncentered* principal components, as is commmon for kernel PCA: we do not subtract the mean before computing the covariance. Since $\Sigma$ is positive semidefinite, it can be diagonalized as

$$\Sigma = U \operatorname{diag}(\boldsymbol{\sigma}^2)U^\top = \sum_i \lambda_i \mathbf{u}_i \mathbf{u}_i^\top$$

where $U = [\mathbf{u}_1, \mathbf{u}_2, \ldots, \mathbf{u}_k]$ is orthonormal and $\operatorname{diag}(\boldsymbol{\sigma}^2) = \operatorname{diag}(\sigma_1^2, \sigma_2^2, \ldots, \sigma_k^2)$ is a diagonal matrix of eigenvalues (here $k = |\mathcal{Z}|$ is the dimension of the feature map). Each of the vectors $\mathbf{u}_i$ is one of the population principal components of the transformed distribution $\phi_+(A)$, giving the directions of maximum variance, and the $\sigma_i^2$ measure the variance in that direction.

Given a new augmentation $a \in \mathcal{A}$, we can then compute the projection of $\phi_+(a)$ into each of these principal component directions as $h_i(a) = \mathbf{u}_i^\top \phi_+(a)$.

**The Markov Chain.** We now redirect our attention to the positive pair Markov chain. The Markov chain transition matrix is defined by $[P_{A \leftarrow A}]_{a_1, a_2} = p_+(a_1 | a_2)$, or in matrix form

$$P_{A \leftarrow A} = P_{A,A} D_A^{-1}.$$

We are interested in the left eigenvectors $\mathbf{f}_i^\top P_{A \leftarrow A} = \lambda_i \mathbf{f}_i^\top$ of this matrix $P_{A \leftarrow A}$, or equivalently the right eigenvectors of its transpose $P_{A \leftarrow A}^\top = P_{A \to A}$, given by $P_{A \to A} \mathbf{f}_i = \lambda_i \mathbf{f}_i$. Observe that then

$$D_A^{-1} P_{A,A} \mathbf{f}_i = \lambda_i \mathbf{f}_i$$

so equivalently

$$D_A^{-1/2} \left( D_A^{-1/2} P_{A,A} D_A^{-1/2} \right) D_A^{1/2} \mathbf{f}_i = \lambda_i D_A^{-1/2} \left( D_A^{1/2} \mathbf{f}_i \right).$$

It follows that $D_A^{1/2} \mathbf{f}_i$ is an eigenvector of the symmetric matrix

$$M = D_A^{-1/2} P_{A,A} D_A^{-1/2}$$

with the same eigenvalue $\lambda_i$. (We note that the matrix $M$ is exactly the symmetrized adjacency matrix described by HaoChen et al. (2021).)

We can now diagonalize $M$ as $M = V \Lambda V^\top$ where $V$ is orthogonal, and then write

$$V = \begin{bmatrix} D_A^{1/2}\mathbf{f}_1 & D_A^{1/2}\mathbf{f}_2 & \ldots & D_A^{1/2}\mathbf{f}_k \end{bmatrix} = D_A^{1/2}[\mathbf{f}_1 \quad \mathbf{f}_2 \quad \ldots \quad \mathbf{f}_k] = D_A^{1/2} F$$

where

$$F = [\mathbf{f}_1 \quad \mathbf{f}_2 \quad \ldots \quad \mathbf{f}_k].$$

Consider an arbitrary function $g : \mathcal{A} \to \mathbb{R}$, and let $\mathbf{g} \in \mathbb{R}^{\mathcal{A}}$ be its vector form, so that $g(a) = \mathbf{g}_a = \mathbf{g}^\top \mathbf{e}_a$. Also define $c_i = \mathbb{E}[g(a)f_i(a)]$ and $\mathbf{c} = [c_1 \quad c_2 \quad \ldots \quad c_k]^\top$. Then

$$\mathbf{c} = \mathbb{E}\left[g(a)F^\top \mathbf{e}_a\right] = \mathbb{E}\left[F^\top \mathbf{e}_a \mathbf{e}_a^\top \mathbf{g}\right] = F^\top D_A \mathbf{g} = V^\top D_A^{1/2} \mathbf{g}$$

so we must have

$$\mathbf{g} = D_A^{-1/2} \left(V^\top\right)^{-1} \mathbf{c} = D_A^{-1/2} V \mathbf{c} = F\mathbf{c}$$

and thus $g(a) = \sum_i c_i f_i(a)$. Additionally, we see that

$$\mathbb{E}\left[g(a)^2\right] = \mathbb{E}\left[\mathbf{g}^\top \mathbf{e}_a \mathbf{e}_a^\top \mathbf{g}\right] = \mathbf{g}^\top D_A \mathbf{g} = \mathbf{c}^\top F^\top D_A F \mathbf{c}$$
$$= \mathbf{c}^\top (D_A^{1/2} F)^\top (D_A^{1/2} F)\mathbf{c} = \mathbf{c}^\top V^\top V \mathbf{c} = \mathbf{c}^\top \mathbf{c} = \sum_i c_i^2.$$

Note that this also implies that the functions $f_i$ are orthonormal under the base measure $p(a)$, e.g. $\mathbb{E}[f_i(a)^2] = 1$ and $\mathbb{E}[f_i(a)f_j(a)] = 0$ for $i \neq j$.

We can now prove our main results from Section 3.

**Proposition 3.1.** *If $g : \mathcal{A} \to \mathbb{R}$ and $c_i = \mathbb{E}[f_i(a)g(a)]$, then*

$$\mathbb{E}_{p_+(a_1,a_2)}\left[\left(g(a_1) - g(a_2)\right)^2\right] = \sum_i (2 - 2\lambda_i)c_i^2. \tag{3}$$

*Proof.* Observe that

$$\mathbb{E}_{p_+}\left[\left(g(a_1) - g(a_2)\right)^2\right] = \mathbb{E}_{p_+}\left[g(a_1)^2 - 2g(a_1)g(a_2) + g(a_2)^2\right]$$
$$= 2\mathbb{E}\left[g(a)^2\right] - 2\mathbb{E}_{p_+}\left[g(a_1)g(a_2)\right]$$
$$= 2\sum_i c_i^2 - 2\mathbb{E}_{p_+}\left[g(a_1)g(a_2)\right]$$

Expanding the second term, we have

$$\mathbb{E}_{p_+}\left[g(a_1)g(a_2)\right] = \mathbb{E}_{p_+}\left[\mathbf{g}^\top \mathbf{e}_{a_1} \mathbf{e}_{a_2}^\top \mathbf{g}\right] = \mathbf{g}^\top \mathbb{E}_{p_+}\left[\mathbf{e}_{a_1} \mathbf{e}_{a_2}^\top\right] \mathbf{g}$$
$$= \mathbf{g}^\top P_{A,A} \mathbf{g}$$
$$= \mathbf{g}^\top D_A^{1/2} M D_A^{1/2} \mathbf{g}$$
$$= \mathbf{g}^\top D_A^{1/2} V \Lambda V^\top D_A^{1/2} \mathbf{g}$$
$$= \mathbf{c}^\top F^\top D_A^{1/2} V \Lambda V^\top D_A^{1/2} F \mathbf{c}$$
$$= \mathbf{c}^\top (D_A^{1/2} F)^\top V \Lambda V^\top (D_A^{1/2} F)\mathbf{c}$$
$$= \mathbf{c}^\top V^\top V \Lambda V^\top V \mathbf{c}$$
$$= \mathbf{c}^\top \Lambda \mathbf{c} = \sum_i \lambda_i c_i^2.$$

We conclude that $\mathbb{E}_{p_+}\left[\left(g(a_1) - g(a_2)\right)^2\right] = 2\sum_i (1 - \lambda_i)c_i^2.$ $\qquad\square$

**Theorem 3.2.** *The output $(h_1, \sigma_1^2), (h_2, \sigma_2^2), \ldots$ of population-level Kernel PCA under $K_+$ and the orthonormal basis of eigenfunctions $f_i$ of $P$ with eigenvalues $\lambda_i$ satisfy $\sigma_i^2 = \lambda_i$ and $h_i(a) = \sigma_i f_i(a) = \lambda_i^{1/2} f_i(a)$ for all $i$ and all $a \in \mathcal{A}$ (up to reordering and multiplicity of eigenspaces[5]).*

*Proof.* Consider the matrix $B = \Phi_+ D_A^{1/2}$, where $\Phi_+ = D_Z^{1/2} P_{Z \to A} D_A^{-1}$ described above. Take the singular value decomposition $B = U\Lambda^{1/2}V^\top$, where $U$ and $V$ are orthonormal and $\Lambda^{1/2}$ is diagonal. Now observe that

$$BB^\top = U\Lambda U^\top = \Phi_+ D_A \Phi_+^\top = \Sigma,$$

and

$$B^\top B = V\Lambda V^\top = D_A^{1/2} \Phi_+^\top \Phi_+ D_A^{1/2} = D_A^{1/2} K_+ D_A^{1/2}$$
$$= D_A^{1/2} \left(D_A^{-1} P_{A,A} D_A^{-1}\right) D_A^{1/2}$$
$$= D_A^{-1/2} P_{A,A} D_A^{-1/2} = M.$$

---

[5]In other words, when some eigenvalues have multiplicity $> 1$, the $h_i$ and $f_i$ are not uniquely determined, but we are free to choose them such that they satisfy this relationship.

Thus, $\Sigma$ and $M$ must have the same eigenvalues. Diagonalize $\Sigma$ and $M$ in terms of $U$ and $V$ and define $h_i(a)$ and $f_i(a)$ according to that diagonalization. We then have

$$
\begin{aligned}
h_i(a) &= \mathbf{u}_i^\top \phi_+(a) \\
&= \mathbf{e}_i^\top U^\top \Phi_+ \mathbf{e}_a \\
&= \mathbf{e}_i^\top U^\top \left( B D_A^{-1/2} \right) \mathbf{e}_a \\
&= \mathbf{e}_i^\top U^\top \left( U \Lambda^{1/2} V^\top D_A^{-1/2} \right) \mathbf{e}_a \\
&= \mathbf{e}_i^\top \Lambda^{1/2} V^\top D_A^{-1/2} \mathbf{e}_a \\
&= \mathbf{e}_i^\top \Lambda^{1/2} \left( D_A^{-1/2} V \right)^\top \mathbf{e}_a \\
&= \mathbf{e}_i^\top \Lambda^{1/2} F^\top \mathbf{e}_a \\
&= \mathbf{e}_i^\top \Lambda^{1/2} [\mathbf{f}_1 \quad \mathbf{f}_2 \quad \ldots \quad \mathbf{f}_k]^\top \mathbf{e}_a \\
&= \lambda_i^{1/2} \mathbf{f}_i^\top \mathbf{e}_a = \lambda_i^{1/2} f_i(a). \qquad \square
\end{aligned}
$$

One interesting consequence of this relationship is that the positive-pair kernel is fully determined by the eigenfunctions and their eigenvalues, and we can write the kernel function directly as a weighted dot product of this representation:

**Proposition C.1.** *For any $a_1, a_2 \in \mathcal{A}$, we have*

$$
K_+(a_1, a_2) = \sum_i \lambda_i f_i(a_1) f_i(a_2),
$$

*where the $f_i$ and $\lambda_i$ are as defined in Section 3.*

*Proof.* Using the matrix notation and definitions described above, algebraic manipulation shows that

$$
\begin{aligned}
K_+(a_1, a_2) &= \mathbf{e}_{a_1} K_+ \mathbf{e}_{a_2} \\
&= \mathbf{e}_{a_1} \left( D_A^{-1} P_{A,A} D_A^{-1} \right) \mathbf{e}_{a_2} \\
&= \mathbf{e}_{a_1} D_A^{-1/2} M D_A^{-1/2} \mathbf{e}_{a_2} \\
&= \mathbf{e}_{a_1} D_A^{-1/2} V \Lambda V^T D_A^{-1/2} \mathbf{e}_{a_2} \\
&= \mathbf{e}_{a_1} D_A^{-1/2} (D_A^{1/2} F) \Lambda (D_A^{1/2} F)^T D_A^{-1/2} \mathbf{e}_{a_2} \\
&= \mathbf{e}_{a_1} F \Lambda F^T \mathbf{e}_{a_2} \\
&= \sum_i \lambda_i f_i(a_1) f_i(a_2). \qquad \square
\end{aligned}
$$

### C.3    RELATIONSHIP TO THE EIGENVECTORS OF THE SYMMETRIZED ADJACENCY MATRIX

Interestingly, the matrix $M$ described above is exactly the symmetrized adjacency matrix discussed by HaoChen et al. (2021). HaoChen et al. motivate their loss as estimating the eigenvectors of $M$ up to a scaling term by $p(a)^{1/2}$, due to prior work showing that eigenvalues give information about clustering structure in graphs.

The connection between the symmetrized adjacency matrix $M$ and the positive-pair Markov chainis well known; indeed, HaoChen et al. briefly discuss the positive-pair Markov chain in their Section 2, and Levin & Peres (2017, Chapter 12) introduce the matrix $M$ when discussing the spectral decomposition of a general symmetric Markov chain.

One way of thinking about this reweighting is as a *change of measure*. The eigenvectors of $M$ are orthonormal with respect to the counting measure over $\mathcal{A}$, e.g. if you sum squared values over all of $\mathcal{A}$, you obtain 1, and the dot product of different eigenvectors is zero. On the other hand, the eigenvectors $\mathbf{f}_i$ (or, equivalently, the eigenfunctions $f_i$) of the Markov chain are orthonormal with respect to the measure $p(A)$, e.g. if you take the expectation of squared values over random augmentations, you obtain 1, and the uncentered covariance of different eigenfunctions is zero.

We believe that using $p(A)$ as a measure is a natural choice, since it allows us to reason about expected values in a straightforward way. From this perspective, the $p(a)^{1/2}$ scaling terms are a desirable feature of the learned representation that allow us to directly reason about optimality with respect to Assumption 1.1.

We note that our Assumption 1.1 could alternatively be expressed in terms of the probability-weighted Laplacian matrix of the augmentation graph, given by $L = D_A - P_{A,A}$. Indeed, we have

$$\mathbb{E}_{p_+}\left[\left(g(a_1) - g(a_2)\right)^2\right] = 2\mathbf{g}^\top L\mathbf{g}.$$

### C.4 RECOVERING PROPORTIONALITY CONSTANTS

As described in Appendix B.1, minimizing the NT-XEnt / InfoNCE loss may not exactly produce the positive-pair kernel $K_+$, but may instead learn a scaled version

$$\widehat{K}_*(a_1, a_2) = \left\langle \sqrt{z(a_1)} \cdot \phi_+(a_1), \ \sqrt{z(a_2)} \cdot \phi_+(a_2) \right\rangle = \sqrt{z(a_1)}\sqrt{z(a_2)}K_+(a_1, a_2),$$

where $z : \mathcal{A} \to \mathbb{R}^+$ is some function which is constant on each communicating class on the Markov chain. (Since $K_+(a_1, a_2) = 0$ whenever $a_1$ and $a_2$ are in separate communicating classes, we could equivalently say $\widehat{K}_*(a_1, a_2) = z(a_1)K_+(a_1, a_2) = z(a_2)K_+(a_1, a_2)$.)

When the Markov chain has one communicating class, $\widehat{K}_*$ is simply a scaled version of $K_+$. In this case, all of the principal component projection functions for $\widehat{K}_*$ are still the eigenfunctions of the Markov chain, but the eigenvalues may be scaled by that constant. The true eigenvalues of the eigenfunctions can then be estimated using equation Equation 4, which states that $\mathbb{E}[(f_i(a_1) - f_i(a_2))^2] = 2(1 - \lambda_i)$.

When the Markov chain has multiple communicating classes, we can partition the eigenfunctions so that each eigenfunction is nonzero on a single communicating class. Since the scaling function $z$ acts as a scaling factor for each communicating class, the principal component functions will then be scaled copies of these partitioned eigenfunctions. We can then similarly estimate the true eigenvalues for each of these eigenfunctions using Equation 4.

# D    GENERALIZATION PROPERTIES OF THE EIGENFUNCTION REPRESENTATION

## D.1    MIN-MAX OPTIMALITY OF EIGENFUNCTIONS

We now prove the min-max optimality of the eigenfunctions with respect to their L2 norm. (We note that these results are closely related to the Courant–Fischer–Weyl min-max principle (Bhatia, 2013, Chapter III), which characterizes the eigenvalues and eigenvectors of a Hermitian matrix in terms of a similar adversarial game.)

**Theorem 4.1.** *Let $\mathcal{F}_r = \{a \mapsto \beta^\top r(a) : \beta \in \mathbb{R}^d\}$ be the subspace of linear predictors from representation $r$, and $S_\varepsilon$ be the set of functions satisfying Assumption 1.1. Let $r_*^d(a) = [f_1(a), f_2(a), \ldots, f_d(a)]$ be the representation consisting of the $d$ eigenfunctions of the positive pair Markov chain with the largest eigenvalues. Then $\mathcal{F}_{r_*^d}$ maximizes the view invariance of the least-invariant unit-norm predictor in $\mathcal{F}_{r_*^d}$:*

$$\mathcal{F}_{r_*^d} = \operatorname*{argmin}_{\dim(\mathcal{F})=d} \max_{\hat{g} \in \mathcal{F}, \; \mathbb{E}[\hat{g}(a)^2]=1} \mathbb{E}_{p_+}\left[\left(\hat{g}(a_1) - \hat{g}(a_2)\right)^2\right]. \tag{5}$$

*Simultaneously, $\mathcal{F}_{r_*^d}$ minimizes the (quadratic) approximation error for the worst-case target function satisfying Assumption 1.1 for any fixed $\varepsilon$:*

$$\mathcal{F}_{r_*^d} = \operatorname*{argmin}_{\dim(\mathcal{F})=d} \max_{g \in S_\varepsilon} \min_{\hat{g} \in \mathcal{F}} \mathbb{E}_{p(a)}\left[\left(g(a) - \hat{g}(a)\right)^2\right]. \tag{6}$$

*Proof.* We will start by deriving Equation 6, and derive Equation 5 afterward. We can think of Equation 6 as equivalent to the following adversarial game:

1. Player chooses a dimension-$d$ subspace $\mathcal{F} \subset \mathcal{A} \to \mathbb{R}$ of functions.

2. Adversary chooses a function $g \in \mathcal{A} \to \mathbb{R}$ with a fixed level of invariance $\mathbb{E}[(g(a_1) - g(a_2))^2] = 2g^\top(D_A - P_{A,A})g = \epsilon$. Without loss of generality, we let $\epsilon = 2$ so that $g^\top(D_A - P_{A,A})g = 1$; other values of $\epsilon$ will just lead to scaling the function $g$.

3. Player chooses the best $\hat{g} \in \mathcal{F}$ to minimize $\mathbb{E}[(\hat{g}(a) - g(a))^2]$

We can analyze this game by working backward from the innermost step, step 3. Given the function class $\mathcal{F}$ and adversarially chosen target function $g$, choosing $\hat{g}$ to minimize the expected squared error is equivalent to finding the orthogonal projection of $g$ into $\mathcal{F}$ with respect to the measure $p(A)$, e.g. with respect to the weighted L2 norm $\mathcal{L}(2; p(A))$. More precisely, we want

$$\hat{g} = \operatorname*{argmin}_{\hat{g} \in \mathcal{F}} \mathbb{E}[(\hat{g}(a) - g(a))^2] = \operatorname*{argmin}_{\hat{g} \in \mathcal{F}}(\hat{\mathbf{g}} - \mathbf{g})^\top D_A(\hat{\mathbf{g}} - \mathbf{g})$$

$$= \operatorname*{argmin}_{\hat{g} \in \mathcal{F}}(D_A^{1/2}\hat{\mathbf{g}} - D_A^{1/2}\mathbf{g})^\top(D_A^{1/2}\hat{\mathbf{g}} - D_A^{1/2}\mathbf{g})$$

But this is just finding the $\hat{\mathbf{g}} \in \mathcal{F}$ which minimizes $\|D_A^{1/2}\hat{\mathbf{g}} - D_A^{1/2}\mathbf{g}\|_2$. This is given by the orthogonal projection of the vector $D_A^{1/2}\mathbf{g}$ into $D_A^{1/2}\mathcal{F}$, under the ordinary L2 norm.

We can now define $R$ as the orthogonal projection operator on $D_A^{1/2}\mathcal{F}$, such that $R\mathbf{h} \in D_A^{1/2}\mathcal{F}$ (e.g. $D_A^{-1/2}R\mathbf{h} \in \mathcal{F}$), and for $D_A^{1/2}\mathbf{f} \in D_A^{1/2}\mathcal{F}$ (e.g. $\mathbf{f} \in \mathcal{F}$), we have $D_A^{1/2}\mathbf{f} = RD_A^{1/2}\mathbf{f}$ (e.g. $\mathbf{f} = D_A^{-1/2}RD_A^{1/2}\mathbf{f}$). Observe that $R$ is real and symmetric, and has eigenvalue 1 with multiplicity $d$ and all other eigenvalues are 0. Since $R$ characterizes the subset, we will find it convenient to redefine our objective for the initial player as choosing $R$, and then letting $\mathcal{F} = \{D_A^{-1/2}RD_A^{1/2}g : g \in \mathcal{A} \to \mathbb{R}\}$.

We then have

$$\hat{\mathbf{g}} = \operatorname*{argmin}_{\hat{\mathbf{g}} \in \mathcal{F}} \mathbb{E}[(\hat{g}(v) - g(v))^2] = D_A^{-1/2}RD_A^{1/2}\mathbf{g}.$$

and the cost is

$$
\begin{aligned}
\mathbb{E}[(\hat{g}(v) - g(v))^2] &= (D_A^{1/2} f - D_A^{1/2} \mathbf{g})^\top (D_A^{1/2} f - D_A^{1/2} \mathbf{g}) \\
&= (R D_A^{1/2} \mathbf{g} - D_A^{1/2} \mathbf{g})^\top (R D_A^{1/2} \mathbf{g} - D_A^{1/2} \mathbf{g}) \\
&= (D_A^{1/2} \mathbf{g})^\top (R - I)^\top (R - I)(D_A^{1/2} \mathbf{g}) \\
&= \mathbf{g}^\top D_A^{1/2} (R^\top R - 2R + I) D_A^{1/2} \mathbf{g} \\
&= \mathbf{g}^\top D_A^{1/2} (I - R) D_A^{1/2} \mathbf{g}.
\end{aligned}
$$

We next consider step 2. Given $R$, what $g$ should the adversary pick? Letting $L = D_A - P_{A,A}$, the adversary is constrained to pick $\mathbf{g}$ such that $\mathbf{g}^\top L \mathbf{g} = (L^{1/2} \mathbf{g})^\top (L^{1/2} \mathbf{g}) = 1$. We note that $L$ is not full rank: in particular, any eigenvector of $M$ with eigenvalue 1 is an eigenvector of $L$ of eigenvector zero. Any function $\mathbf{g}$ chosen by the adversary must then be the sum of two parts:

- a component in in the range of $L$, of the form $(L^\dagger)^{1/2} \mathbf{u}$ where $\|\mathbf{u}\|_2 = 1$ and † represents the Moore-Penrose pseudoinverse,

- and a component in the null space of $L$.

Overall, we can thus write $\mathbf{g} = (L^\dagger)^{1/2} \mathbf{u} + \mathbf{h}$ where $\|\mathbf{u}\|_2 = 1$ and $\mathbf{h}^\top L \mathbf{h} = 0$. Similarly, the response $\hat{\mathbf{g}}$ must also have two components, one in the range of $L$ and one in the null space of $L$. There are then two cases. If $\mathcal{F}$ does not span the entire null space of $L$, the adversary can force an arbitrarily high approximation error by choosing $\mathbf{h}$ to be in the null space of $L$ but not $\mathcal{F}$. On the other hand, if $\mathcal{F}$ spans the entire null space of $L$, the player can always perfectly approximate $\mathbf{h}$, and so the adversary is forced to maximize cost by using $\mathbf{u}$. In particular, they will pick

$$
\begin{aligned}
\mathbf{u} &= \underset{\|\mathbf{u}\|_2 = 1}{\arg\max} ((L^\dagger)^{1/2} \mathbf{u})^\top D_A^{1/2} (I - R) D_A^{1/2} ((L^\dagger)^{1/2} \mathbf{u}) \\
&= \underset{\|\mathbf{u}\|_2 = 1}{\arg\max} \, \mathbf{u}^\top (L^\dagger)^{1/2} D_A^{1/2} (I - R) D_A^{1/2} (L^\dagger)^{1/2} \mathbf{u} \\
&= \underset{\|\mathbf{u}\|_2 = 1}{\arg\max} \, \mathbf{u}^\top A \mathbf{u}
\end{aligned}
$$

where $A$ is the matrix $(L^\dagger)^{1/2} D_A^{1/2} (I - R) D_A^{1/2} (L^\dagger)^{1/2}$. The optimal choice for $\mathbf{u}$ is an eigenvector of $A$ with maximal eigenvalue, and the cost is then that maximal eigenvalue. But observe that $M$ is similar to the following:

$$
\begin{aligned}
A &\sim ((L^\dagger)^{1/2} D_A^{1/2})^{-1} M ((L^\dagger)^{1/2} D_A^{1/2}) \\
&= (I - R) D_A^{1/2} L^\dagger D_A^{1/2} \\
&= (I - R) \left( D_A^{-1/2} L D_A^{-1/2} \right)^\dagger := A'
\end{aligned}
$$

Similar matrices have the same eigenvalues, so the maximum cost attainable by the adversary is the maximal eigenvalue of $A'$.

Finally, we consider step 1. Which $R$ should our player choose to minimize this maximum cost? They should first ensure the cost is finite, by choosing $R$ to span the null space of $D_A^{-1/2} L D_A^{-1/2}$. (Note that if $d$ is less than the dimension of this null space, there is no choice that ensures a finite cost; in this case every representation has unbounded worst-case approximation error.) Afterward, they should ensure that $A'$ has the smallest maximum eigenvalue. The sorted vector of eigenvalues of $A'$ is bounded below by the vector obtained by matching the largest eigenvalues of $\left( D_A^{-1/2} L D_A^{-1/2} \right)^\dagger$ with the smallest of $(I - R)$ (Bhatia, 2013, exercise III.6.14)[6]. Let $d^*$ be the dimension of the null space of $D_A^{-1/2} L D_A^{-1/2}$. Then $I - R$ has $d$ eigenvalues with value 0 (i.e. 0 is an eigenvalue with multiplicity $d$). Of these, $d^*$ must be used to span this null space, and the remaining $d - d^*$ (if any)

---

[6]See also https://math.stackexchange.com/questions/573583/eigenvalues-of-the-product-of-two-symmetric-matrices

can be used to reduce the eigenvalues of $A'$. Thus, the largest eigenvalue of $A'$ is always at least as big as the $(d - d^* + 1)$-th largest eigenvalue of $\left(D_A^{-1/2} L D_A^{-1/2}\right)^\dagger$. We can attain this bound by setting $R$ to exactly capture the $(d - d^*)$-dimensional subspace of $\left(D_A^{-1/2} L D_A^{-1/2}\right)^\dagger$ spanned by its top eigenspaces, along with the $d^*$-dimensional null space of $D_A^{-1/2} L D_A^{-1/2}$.

But the combination of the null space of $D_A^{-1/2} L D_A^{-1/2}$ and the top eigenspace of $\left(D_A^{-1/2} L D_A^{-1/2}\right)^\dagger$ is just the space spanned by the $d$ eigenvectors of $D_A^{-1/2} L D_A^{-1/2}$ with the *smallest* eigenvalues. Furthermore,

$$D_A^{-1/2} L D_A^{-1/2} = D_A^{-1/2}(D_A - P_{A,A})D_A^{-1/2}$$
$$= I - D_A^{-1/2} P_{A,A} D_A^{-1/2} = I - M,$$

so we are looking for the eigenvectors of $M$ with the *largest* eigenvalues, where $M$ is the matrix described in Appendix C.

Thus, the player should choose $\mathcal{F}$ such that $D_A^{1/2}\mathcal{F}$ spans the top $d$-dimensional eigenspace of $M$, e.g. they should choose functions of the form $D_A^{-1/2}\mathbf{v}_i$ where the $\mathbf{v}_i$ are the eigenvectors of $M$ with largest eigenvalue. But these are exactly the left eigenvectors $\mathbf{f}_i$ of the positive pair Markov chain, which is how $r_*^d$ is defined. We conclude that $\mathcal{F}_{r_*^d}$ is the optimal choice for the player, and thus Equation 6 holds.

Indeed, we can conclude something further: if $\lambda_{d+1}$ is the $(d+1)$th eigenvalue of the positive pair Markov chain (the variance along the $(d+1)$th principal component of the positive-pair kernel), then as long as $d \geq d^*$, $(1 - \lambda_{d+1})^{-1}$ is the $(d - d^* + 1)$th eigenvalue of $\left(D_A^{-1/2} L D_A^{-1/2}\right)^\dagger$, which is exactly the worst-case approximation error for $\mathcal{F}_{r_*^d}$ against any function with $\mathbb{E}[(g(v_1) - g(v_2))^2] = 2g^\top L g = 2$. Scaling by $\varepsilon$, if $\mathbb{E}[(g(v_1) - g(v_2))^2] = \varepsilon$ then the worst case error is $\frac{1}{2}\varepsilon/(1 - \lambda_{d+1})$. In other words,

$$\max_{g \in S_\epsilon} \min_{\hat{g} \in \mathcal{F}_{r_*^d}} \mathbb{E}_{p(a)}\left[\left(g(a) - \hat{g}(a)\right)^2\right] = \frac{\varepsilon}{2(1 - \lambda_{d+1})}.$$

We now return our attention to Equation 5. This equation can also be formulated as an adversarial game:

1. Player chooses a rank-$d$ subspace $\mathcal{F} \subset \mathcal{A} \to \mathbb{R}$ of functions.

2. Adversary chooses a function $\hat{g} \in \mathcal{F}$ with unit norm $\mathbb{E}[\hat{g}(v)^2] = \hat{\mathbf{g}}^\top D_A \hat{\mathbf{g}} = 1$ to maximize $\mathbb{E}[(\hat{g}(v_1) - \hat{g}(v_2))^2] = 2\hat{\mathbf{g}}^\top L \hat{\mathbf{g}}$.

We can again identify the choice of $\mathcal{F}$ with the choice of the orthogonal projection matrix $R$ on $D_A^{1/2}\mathcal{F}$. We know $\hat{g} \in \mathcal{F}$, so we can write $\hat{\mathbf{g}} = D_A^{-1/2} R D_A^{1/2}\hat{\mathbf{g}}$. Also note that for any $\mathbf{h}$ (not even necessarily in $\mathcal{F}$), $D_A^{-1/2} R D_A^{1/2}\mathbf{h} \in \mathcal{F}$. Now suppose we choose an $\mathbf{h}$ so that $\mathbb{E}[h(a)^2] = \mathbf{h}^\top D_A \mathbf{h} = 1$, and define $\hat{\mathbf{g}} = D_A^{-1/2} R D_A^{1/2}\mathbf{h}$. Then

$$\hat{\mathbf{g}}^\top D_A \hat{\mathbf{g}} = \mathbf{h}^\top D_A^{1/2} R D_A^{-1/2} D_A (D_A^{-1/2} R D_A^{1/2}\mathbf{h})$$
$$= \mathbf{h}^\top D_A^{1/2} R^2 D_A^{1/2}\mathbf{h}$$
$$= \mathbf{h}^\top D_A^{1/2} R D_A^{1/2}\mathbf{h}$$
$$\leq \mathbf{h}^\top D_A^{1/2} I D_A^{1/2}\mathbf{h} = 1$$

because $R$ has eigenvalues at most 1. So, the following are equivalent:

- choosing $\hat{\mathbf{g}} \in \mathcal{F}$ with $\mathbb{E}[\hat{g}(v)^2] \leq 1$

- choosing $\mathbf{h} \in \mathbb{R}^{\mathcal{A}}$ with $\mathbb{E}[h(v)^2] \leq 1$ and letting $\hat{\mathbf{g}} = D_A^{-1/2} R D_A^{1/2}\mathbf{h}$

We also note that there is no advantage to the adversary from picking a function such that $\mathbb{E}[\hat{g}(v)^2] < 1$. So we can reframe step 2 as choosing $\mathbf{h}$ so that $\mathbb{E}[h(v)^2] = 1$, to maximize

$$2\left(D_A^{-1/2}RD_A^{1/2}\mathbf{h}\right)^\top L\left(D_A^{-1/2}RD_A^{1/2}\mathbf{h}\right)$$

We can further reparameterize by letting $\mathbf{h} = D_A^{-1/2}\mathbf{u}$, so that $\mathbb{E}[h(v)^2] = 1$ is equivalent to $\|\mathbf{u}\|_2^2 = 1$. We then have cost

$$\begin{aligned}
C = 2\hat{\mathbf{g}}^\top L\hat{\mathbf{g}} &= 2(\mathbf{h}^\top D_A^{1/2}RD_A^{-1/2})L(D_A^{-1/2}RD_A^{1/2}\mathbf{h}) \\
&= 2(\mathbf{u}^\top D_A^{-1/2})D_A^{1/2}RD_A^{-1/2}LD_A^{-1/2}RD_A^{1/2}(D_A^{-1/2}\mathbf{u}) \\
&= 2\mathbf{u}^\top RD_A^{-1/2}LD_A^{-1/2}R\mathbf{u}
\end{aligned}$$

The choice that maximizes the cost is then an eigenvector of

$$B = 2RD_A^{-1/2}LD_A^{-1/2}R = \left(L^{1/2}D_A^{-1/2}R\right)^\top \left(L^{1/2}D_A^{-1/2}R\right)$$

with maximal eigenvalue, and the cost is 2 times the maximum eigenvalue of $B$. But note that $B$ has the same eigenvalues as

$$B' = \left(L^{1/2}D_A^{-1/2}R\right)\left(L^{1/2}D_A^{-1/2}R\right)^\top = L^{1/2}D_A^{-1/2}RD_A^{-1/2}L^{1/2}$$

since $R^2 = R$. And $B'$ is similar to

$$B'' = D_A^{-1/2}LD_A^{-1/2}R$$

so $B$ and $B''$ have the same eigenvalues.

We now consider step 1. What should the player choose for $R$? By a similar eigenvalue-of-product argument as used for Equation 6, regardless of the choice of $R$ the largest eigenvalue of $B''$ must always be at least as big as the $d$th smallest eigenvalue of $D_A^{-1/2}LD_A^{-1/2}$, because $R$ has eigenvalue 1 with multiplicity $d$. We can attain this minimum cost by choosing $R$ to project into the eigenspace spanned by the $d$ eigenvectors of $D_A^{-1/2}LD_A^{-1/2}$ with the *smallest* eigenvalues.

But observe that the smallest eigenvalues and corresponding eigenvectors of $D_A^{-1/2}LD_A^{-1/2} = I - M$ are exactly the largest eigenvalues and corresponding eigenvectors of $M = D_A^{-1/2}P_{A,A}D_A^{-1/2} = I - L$, which as we argued above, is exactly the set of eigenfunctions $f_i$ used to construct $r_*^d$.

In this case, the optimal cost itself is determined by the largest eigenvalue of $D_A^{-1/2}LD_A^{-1/2}$ (times two), so we obtain

$$\max_{\substack{\hat{g}\in\mathcal{F}_{r_*^d},\\\mathbb{E}[\hat{g}(a)^2]=1}}\mathbb{E}_{p_+}\left[\left(\hat{g}(a_1) - \hat{g}(a_2)\right)^2\right] = 2(1 - \lambda_d).$$

$\square$

## D.2 GENERALIZATION BOUND FOR LINEAR PREDICTION WITH THE EIGENFUNCTION REPRESENTATION

**Proposition 4.2.** *Let $(A_i, Y_i)_{i=1}^n$ be i.i.d. samples, choose $R \geq 0$, and consider the constrained empirical risk minimizer $\hat{\beta}_R \in \arg\min_{\|\beta\|_2 \leq R} n^{-1}\sum_{i=1}^n |\langle\beta, r_d^*(A_i)\rangle - Y_i|$. Then the expected excess risk of $\hat{\beta}_R$ is bounded by:*

$$\mathbb{E}\left[\mathcal{E}(\hat{\beta}_R)\right] \leq \frac{2dR}{\sqrt{n}} + \sqrt{d}(\|\beta^*\|_2 - R)_+ + \sqrt{\frac{\varepsilon}{2(1 - \lambda_{d+1})}}$$

*where $\mathcal{E}(\beta) := \mathcal{R}(\beta) - \mathcal{R}^*$ is the excess risk and $(x)_+ := \max\{x, 0\}$.*

*Proof.* We start by decomposing the excess risk as:

$$
\mathcal{R}(\hat{\beta}_R) - \mathcal{R}^* = \left( \mathcal{R}(\hat{\beta}_R) - \inf_{\|\beta\|_2 \leq R} \mathcal{R}(\beta) \right) + \left( \inf_{\|\beta\|_2 \leq R} \mathcal{R}(\beta) - \mathcal{R}(\beta^*) \right) + (\mathcal{R}(\beta^*) - \mathcal{R}^*)
$$

The first term is the estimation error, which we can readily bound by first noting that:

$$
\mathbb{E}\big[\|r_*^d(A)\|_2^2\big] = \mathbb{E}\left[\sum_{i=1}^d f_i(A)^2\right] = \sum_{i=1}^d \mathbb{E}\big[f_i(A)^2\big] = d
$$

then noticing that our loss is 1-Lipschitz, and finishing with the standard Rademacher complexity argument for constrained linear classes Kakade et al. (2008) to get:

$$
\left( \mathcal{R}(\hat{\beta}_R) - \inf_{\|\beta\|_2 \leq R} \mathcal{R}(\beta) \right) \leq \frac{2dR}{\sqrt{n}} \tag{7}
$$

The second term is an approximation error term due to the use of a constrained linear class instead of the full linear class. Define $\tilde{\beta}^* := \frac{\beta^*}{\max\{\|\beta^*\|_2/R, 1\}}$. Then we can bound this second term by:

$$
\begin{aligned}
\inf_{\|\beta\|_2 \leq R} \mathcal{R}(\beta) - \mathcal{R}(\beta^*) &\leq \mathcal{R}(\tilde{\beta}^*) - \mathcal{R}(\beta^*) \\
&\leq \mathbb{E}\big[\big|\big\langle r_*^d(A), \tilde{\beta}^* - \beta^*\big\rangle\big|\big] \\
&\leq \mathbb{E}\big[\|r_*^d(A)\|_2\big]\|\tilde{\beta}^* - \beta^*\|_2 \\
&\leq \sqrt{d}(\|\beta^*\|_2 - R)_+
\end{aligned} \tag{8}
$$

where the second inequality follows from the 1-Lipschitznes of the loss, the third by Cauchy-Schwartz inequality, and the last by Jensen's inequality.

The third and last term is an approximation error term due to the use of the function class given by the span of the first $d$ eigenfunctions $(f_i)_{i=1}^d$. We can bound it as follows:

$$
\begin{aligned}
\mathcal{R}(\beta^*) - \mathcal{R}^* &\leq \mathbb{E}\big[\big|\langle r_*^d(A), \beta^*\rangle - g^*(A)\big|\big] \\
&\leq \sqrt{\mathbb{E}[(\langle r_*^d(A), \beta^*\rangle - g^*(A))^2]} \leq \sqrt{\frac{\varepsilon}{2(1-\lambda_{d+1})}}
\end{aligned} \tag{9}
$$

where the first inequality follows from the 1-Lipschitznes of the loss, the second from Jensen's inequality, and the last by first noticing that the function $h(a) := \big\langle r_*^d(a), \beta^*\big\rangle$ satisfies $h = \mathrm{argmin}_{g \in \mathcal{F}_{r_*^d}} \mathbb{E}\big[(g(A) - g^*(A))^2\big]$ (since it is the projection of $g^*$ onto $\mathcal{F}_{r_*^d}$ under the norm $\|x\|^2 := \mathbb{E}\big[x^2(A)\big]$), then appealing to the proof of Proposition 4.1. Combining the bounds of equations (7), (8), and (9), we obtain the stated generalization bound. $\qquad\square$

## E    EIGENFUNCTION ESTIMATION TECHNIQUES

In practice, we do not generally have access to the closed form for $K_+$ or the the full population of our examples, but instead only have access to a dataset of positive pairs $(a_1, a_2)$ drawn from the distribution $p_+(a_1, a_2)$ (or, more commonly, a dataset of examples $z$ and a sampling algorithm for $p(A|Z)$). In this section we discuss some approaches for approximating the optimal eigenfunction representation from these samples.

### E.1    COMBINING CONTRASTIVE LEARNING WITH KERNEL PCA

Our analysis in Sections 2 and 3 motivates the following procedure:

1. Train a contrastive learning model using an existing contrastive learning objective.

2. Using the equations in Table 1, identify the approximate kernel $\widehat{K}_\theta$, which will hopefully be similar to the positive-pair kernel $K_+$ assuming we have converged to a solution to the objective.

3. Perform (or approximate) Kernel PCA using $\widehat{K}_\theta$ and a large set of individual views drawn from $p(A)$.

4. Use the first $d$ extracted principal component projection functions $h_i : \mathcal{A} \to \mathbb{R}$ to construct a representation, possibly normalizing by $\sigma_i$ to obtain the orthonormal basis $f_i(a) = \sigma_i^{-1} h_i(a)$.

We note that applying a rotation matrix to the optimal representation does not affect the expressivity of that representation. It is thus sufficient to identify the subspace spanned by the first $d$ principal component projection functions. If the representation dimension $d$ is known in advance, it may be possible to adjust the contrastive learning method to accomplish this without requiring a separate PCA step. In particular, when using the spectral contrastive loss with a $d$-dimensional linear kernel head, the population minimizer of the loss will exactly correspond to the best $d$-dimensional approximation of $K_+$. This means that the output layer representation will be exactly the set of principal component projection functions rotated by some orthogonal matrix.

On the other hand, including the PCA step makes it possible to decouple the representation dimension from the kernel approximation method, which may be advantageous if the learning dynamics of a different parameterization or loss are better, or if $d$ is not known in advance. The PCA step also makes it possible to diagnose how well the learned representation is capturing properties of $K_+$ by checking the extent to which Equation 4 is satisfied.

Directly applying kernel PCA can be expensive for large datasets, due to the need to decompose the full Gram matrix of kernel similarities. A more computationally-friendly approach is to first approximate $\widehat{K}_\theta$ with a lower-rank approximation, such as the Nyström method (Williams & Seeger, 2000), and then perform PCA over that approximation (Sriperumbudur & Sterge, 2017; Ullah et al., 2018; Sterge et al., 2020). This can be especially useful when the dataset is much larger than the number of desired eigenfunctions.

We note that approaches based on Kernel PCA may be numerically unstable in the presence of many eigenvalues close to 1, since small kernel estimation errors may be amplified by the eigendecomposition process. Although we were able to apply these techniques to models trained on our two synthetic datasets, we have been so far unable to obtain a reliable estimate of the eigenfunctions for real-world datasets such as those considered by Chen et al. (2020a). In particular, we attempted to apply the Nyström method to a pretrained SimCLR model but ran into numerical precision issues and were unable to form a good approximation of the learned hypersphere-based kernel $\widehat{K}_\theta$. See also Appendix F.5 for a preliminary analysis of a model with a constrained-norm linear kernel head on ImageNet; although we were able to run Kernel PCA with this model, it does not appear to be a good approximation of $K_+$.

### E.2 DIRECT EIGENFUNCTION EXTRACTION, AND CONNECTIONS TO VICREG

An alternative strategy for estimating the eigenfunctions $f_i$ is to combine the contrastive learning and Kernel PCA steps into a single parameterized model. This is possible using parameterized spectral decomposition techniques such as SpIN (Pfau et al., 2018) or NeuralEF (Deng et al., 2022).

We note that these techniques are usually motivated as extracting the eigenfunctions of the kernel operator $T[f](a_1) = \int K(a_1, a_2) f(a_2) p(a_2) \, da_2$, or in other words, solving the eigenfunction equation

$$\lambda_i f_i(a_1) = \int K(a_1, a_2) f_i(a_2) p(a_2) \, da_2.$$

In this case, substituting the form of $K_+$ reveals that this is equivalent to finding the eigenfunctions of the positive pair Markov chain:

$$\lambda_i f_i(a_1) = \sum_{a_2} K_+(a_1, a_2) f_i(a_2) p(a_2) = \sum_{a_2} \frac{p_+(a_1, a_2)}{p(a_1)p(a_2)} f_i(a_2) p(a_2)$$

$$= \sum_{a_2} \frac{p_+(a_1, a_2)}{p(a_1)} f_i(a_2)$$

$$= \sum_{a_2} p(a_2|a_1) f_i(a_2)$$

$$= \mathbb{E}_{a_2|a_1}[f_i(a_2)]$$

**Connections between NeuralEF and VICReg.**  We now describe in more detail how to apply the NeuralEF technique to estimate the basis of eigenfunctions of the positive pair Markov chain. The NeuralEF approach approximates the eigenfunctions $f_i$ of a kernel $K$ by solving an asymmetric set of constrained optimization problems $\hat{f}_j = \mathrm{argmax}_{\hat{f}_j} R_{jj} - \sum_{i=0}^{j-1} \frac{R_{ij}^2}{R_{ii}}$ subject to the constraint that $C_j = 1$, where

$$R_{ij} = \sum_{a_1,\, a_2} \hat{f}_i(a_1) K(a_1, a_2) \hat{f}_j(a_2) p(a_1) p(a_2), \qquad C_j = \sum_a \hat{f}_j(a)^2 p(a) = \mathbb{E}\left[ \hat{f}_j(a)^2 \right].$$

Substituting the positive-pair kernel $K_+(a_1, a_2) = \frac{p_+(a_1,a_2)}{p(a_1)p(a_2)}$ reveals an alternative form for the $R_{ij}$ terms, allowing us to apply NeuralEF using samples from $p_+$:

$$R_{ij} = \sum_{a_1,\, a_2} \hat{f}_i(a_1) \frac{p_+(a_1, a_2)}{p(a_1)p(a_2)} \hat{f}_j(a_2) p(a_1) p(a_2) = \mathbb{E}_{p_+(a_1,a_2)}\left[ \hat{f}_i(a_1) \hat{f}_j(a_2) \right].$$

Interestingly, the resulting algorithm closely resembles the Variance-Invariance-Covariance regularization (VICReg) self-supervised learning method proposed by Bardes et al. (2021): the $C_j = 1$ constraint ensures each function has sufficient variance, the $R_{ij}^2$ term reduces the covariance between features, and maximizing $R_{jj} = \mathbb{E}[\hat{f}_j(a_1) \hat{f}_j(a_2)]$ over positive pairs leads to representations that are as invariant as possible between positive pairs. (Note, however, that the asymmetric weighted covariance penalties $\frac{R_{ij}^2}{R_{ii}}$ in NeuralEF ensure that eigenfunctions are recovered in the correct order without interfering with one another.)

**Stabilizing NeuralEF for contrastive learning.**  Althoug the NeuralEF-based approach works well when $R_{ii}$ is large for all $i$, the method becomes numerically unstable when $R_{ii}$ is small. And since $R_{ii} \approx \lambda_i$, this makes it difficult to recover eigenfunctions whose eigenvalues $\lambda_i$ are close to zero.

To enable recovery of all eigenfunctions, including those with $\lambda_i = 0$, we do not directly apply Neural EF to $K_+$ in our experiments. Instead, we construct a modified kernel with the help of a modified positive pair distribution

$$p_{\mathrm{mix}}(a_1, a_2) = 0.5 p_+(a_1, a_2) + 0.5 p(a_1) \mathbb{1}[a_2 = a_1]$$

where $\mathbb{1}[a_2 = a_1]$ is 1 when $a_1 = a_2$ and 0 otherwise. Conceptually, with probability 50%, we sample a positive pair $(a_1, a_2)$ as normal, and otherwise, we sample a single augmentation $a_1$ and then choose $a_2 = a_1$.

The corresponding positive-pair Markov chain transition matrix can be written as

$$P_{\mathrm{mix}} = 0.5P + 0.5I.$$

It follows that the eigenfunctions for this modified distribution are the same as those for our original positive pair distribution, but the eigenvalues are transformed according to $\lambda_i^{\mathrm{mix}} = 0.5\lambda_i + 0.5$. We can thus apply Neural EF to the corresponding kernel $K_+^{\mathrm{mix}}(a_1, a_2) = p_{\mathrm{mix}}(a_1, a_2)/p(a_1)p(a_2)$, and recover the original $\lambda_i$ by inverting this transformation.

Substituting this into the Neural EF objective, we obtain

$$
\begin{aligned}
R_{ij}^{\mathrm{mix}} &= \sum_{a_1,\, a_2} \hat{f}_i(a_1)\frac{p_{\mathrm{mix}}(a_1, a_2)}{p(a_1)p(a_2)}\hat{f}_j(a_2)p(a_1)p(a_2) \\
&= 0.5 \sum_{a_1,\, a_2} \hat{f}_i(a_1)\frac{p_+(a_1, a_2)}{p(a_1)p(a_2)}\hat{f}_j(a_2)p(a_1)p(a_2) \\
&\qquad + 0.5 \sum_{a_1,\, a_2} \hat{f}_i(a_1)\frac{p(a_1)\mathbb{1}[a_2 = a_1]}{p(a_1)p(a_2)}\hat{f}_j(a_2)p(a_1)p(a_2) \\
&= 0.5 \sum_{a_1,\, a_2} \hat{f}_i(a_1)\hat{f}_j(a_2)p_+(a_1, a_2) + 0.5 \sum_{a_1,\, a_2} \hat{f}_i(a_1)\hat{f}_j(a_2)p(a_1)\mathbb{1}[a_2 = a_1] \\
&= 0.5\, \mathbb{E}_{p_+(a_1,a_2)}\left[\hat{f}_i(a_1)\hat{f}_j(a_2)\right] + 0.5\, \mathbb{E}_{p(a_1)}\left[\hat{f}_i(a_1)\hat{f}_j(a_1)\right].
\end{aligned}
$$

We then estimate $R_{ij}^{\mathrm{mix}}$ in each minibatch by averaging over minibatch positive pairs for the first term and over all minibatch augmentations for the second term. (In practice, we drop the 0.5 scaling factor in the loss.) We find that this modification greatly stabilizes the Neural EF objective when estimating large numbers of eigenfunctions, and in particular makes it possible to learn eigenfunctions of $K_+$ with eigenvalues that are very small or even zero.

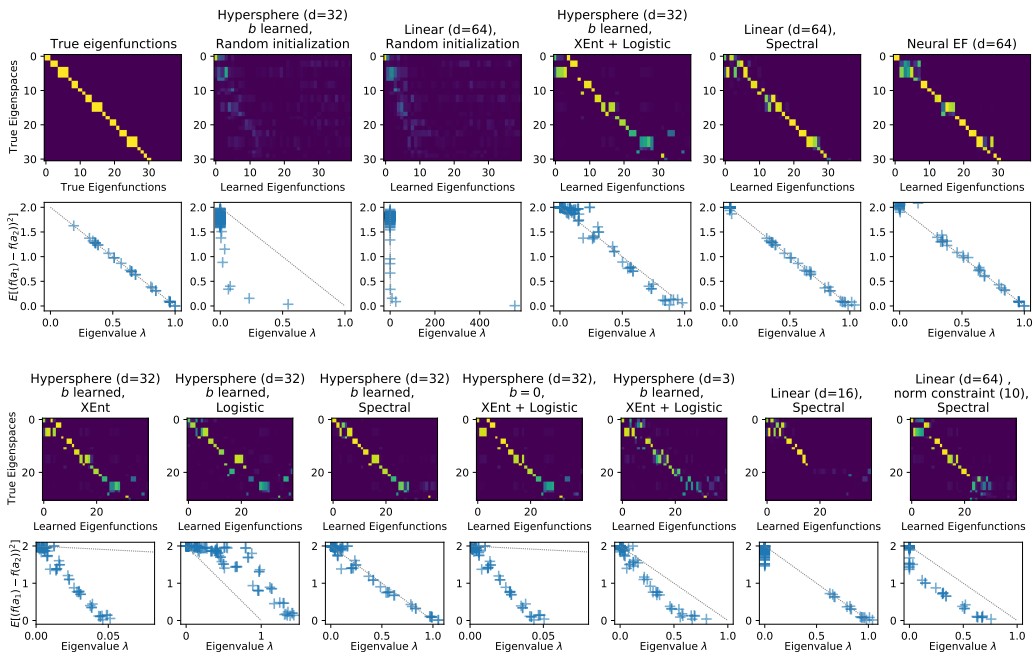

Figure 6: Eigenfunction and eigenvalue estimation accuracy on the overlapping regions task, for a selection of models. Top row: Alignment between true eigenspaces and the learned kernel principal components, with perfect alignment shown as a block diagonal matrix. Bottom row: Relationship between learned kernel eigenvalue $\lambda$ and the corresponding positive-pair discrepancy $\mathbb{E}_{p_+}\left[(f(a_1) - f(a_2))^2\right]$, with the relationship predicted by Equation 4 shown with a dashed line.

# F    EXPERIMENT DETAILS

In this section we describe our experiments and their results in more detail. We start by presenting the full set of results summarized in Section 6 and Figure 5. We then describe details regarding model training. We conclude with some additional preliminary results regarding supervised learning with Kernel PCA representations on MNIST and eigenfunction extraction on ImageNet.

## F.1    ADDITIONAL EIGENFUNCTION ESTIMATION RESULTS

**Overlapping regions task.**    Results for our full set of models on the "overlapping regions" task are shown in Figure 6.

We find that, under suitable losses, *linear kernels, hypersphere kernels, and NeuralEF can all produce good approximations of the basis of eigenfunctions*, despite their diferent parameterizations. Specifically, unconstrained-norm linear kernels with the spectral loss, hypersphere kernels with a learned bias under either a XEnt-Logistic loss mixture or the spectral loss, and the NeuralEF eigenfunction estimation method all produce eigenfunction estimates that are reasonably aligned to the true eigenfunctions, and eigenvalue estimates that are close to the true eigenvalues. However, especially for eigenspaces with similar eigenvalues, the eigenfunctions occasionally appear in the incorrect order, and have a small amount of mixing between eigenspaces. The relationship between approximate eigenvalues and positive-pair discrepancies also closely matches the prediction from Equation 4.

We note also that *alignment with the basis of eigenfunctions emerges during training,* and is not simply a property of the model architecture, as evidenced by the lack of alignment when applying Kernel PCA to randomly initialized models.

*The loss function used influences the learning dynamics and final result.* Using the XEnt loss alone produces a reasonably-well aligned eigenfunction decomposition, but has eigenvalues scaled by a

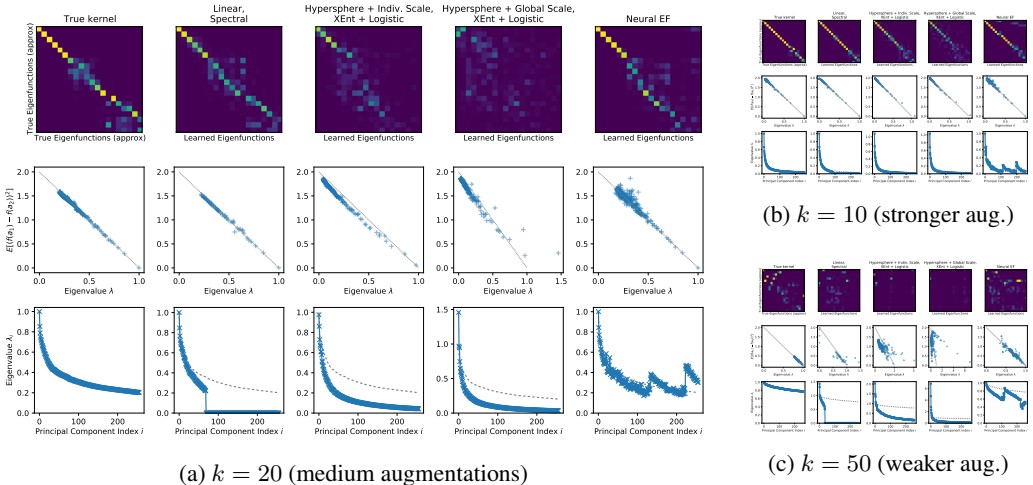

(a) $k = 20$ (medium augmentations)

(b) $k = 10$ (stronger aug.)

(c) $k = 50$ (weaker aug.)

Figure 7: Alignments and discrepancy relationships for principal component functions on MNIST task for three augmentation strengths. Top row: Alignment (squared dot products) of the first 20 principal component functions between runs of Kernel PCA on different kernels, with perfect alignment corresponding to a diagonal matrix. (Left column shows alignment between two independent runs of Kernel PCA on the true kernel $K_+$.) Middle row: Relationship between eigenvalue $\lambda$ and positive-pair discrepancy for the first 256 principal components (omitting any with $\lambda = 0$), with the prediction from Equation 4 shown as a gray dashed line. Bottom row: Ordered eigenvalues for each approximation, relative to those of the true kernel (dashed line).

constant, since the XEnt loss only measures ratios between kernel evaluations and thus only recovers the kernel up to a constant factor (discussed in Appendix B.1). The Logistic loss alone appears to lead to inferior decomposition quality, although eigenvalues are in the right order of magnitude. Interestingly, the spectral loss seems to work even for hypersphere kernel approximations, although we found it to be the most unstable; successfully training a hypersphere kernel with the spectral loss required initializing the bias $b$ to a large negative number.

*Constraints on the kernel approximation degrade eigenfunction and eigenvalue estimates.* For the hypersphere kernel, fixing the bias $b$ to zero led to eigenvalues that were abnormally small, whereas reducing the dimensionality of the hypersphere from 32 to 3 both degraded eigenfunction alignment and led to deviations from Equation 4. For the linear kernel, using a smaller dimension led to estimating only the eigenfunctions with larger eigenvalues, and imposing a norm constraint of $\|h_\theta(a)\|^2 = 10$ both reduced the number of accurately-captured eigenfunctions and caused the eigenvalues to be smaller than predicted by Equation 4.

**MNIST task.**   Results for our full set of models at three augmentation strengths are shown in Figure 7.

We compare two types of hypersphere kernel parameterization: a "global scale" version using $\exp(h_\theta(a_1)^\top h_\theta(a_2)/\tau + b)$ and a "local scale" version using $\exp(h_\theta(a_1)^\top h_\theta(a_2)/\tau + s_\theta(a_1) + s_\theta(a_2))$. The second is more expressive, but the first is closer to that considered by prior work such as Chen et al. (2020a). (Note that most work with hypersphere-based models fixes $b = 0$, but also uses the XEnt loss alone, which is not affected by the value of $b$. We include $b$ and include the Logistic loss to assess how well the models can recover the correct values of the eigenvalues.).

Note that, although we can exactly evaluate $K_+$ on any pair of augmented views, the space $\mathcal{A}$ (containing all multisets of $k$ pixels) is too large to enumerate, preventing us from exactly computing the exact eigenfunctions of $K_+$. Instead, we use Kernel PCA over a large set of samples to estimate the "ground truth" eigenfunctions. To better understand the impact of this step, we also include a comparison between two independent runs of Kernel PCA on $K_+$.

*Weaker augmentations make principal component estimation difficult.* We find that Kernel PCA can more reliably recover principal components with large gaps between eigenvalues, and is influenced

by random noise as eigenvalues become closer together. In particular, as augmentation strength decreases, there are more eigenvalues close to 1, and it becomes more difficult to identify the most significant principal components. Due to the stochasticity of Kernel PCA, it is difficult to accurately identify eigenfunctions even when given direct access to $K_+$, and two runs of Kernel PCA begin to diverge as eigenvalues decrease. Eigenfunction quality degrades even faster when comparing results of Kernel PCA for a learned model and for $K_+$: the learned models only allow recovery of a few principal components accurately at larger augmentation strengths.

*More expressive kernel approximations recover more eigenfunctions.* We observe that the linear kernel head and Neural EF method are able to recover a larger number of eigenfunctions accurately compared to hypersphere kernels, and have eigenvalues that lie closer to the Equation 4 prediction. Additionally, we find that adding a per-example scale function $s_\theta$ to the hypersphere kernel leads to more correctly-recovered eigenfunctions and fewer outlier eigenvalues.

*Learned models have faster eigenvalue decay than $K_+$.* In general, the eigenvalues of learned kernels decay faster than the eigenvalues of the true positive-pair kernel $K_+$. Interestingly, however, the eigenvalues still appear to follow the relationship predicted by Equation 4 for sufficiently expressive models and sufficiently strong augmentations. This suggests that the learned models are approximately capturing a subset of the positive-pair eigenfunctions.

We note that both the linear-kernel-head model and the NeuralEF model exhibit a sharp change in eigenvalue near the 100th eigenfunction: the first shows a sudden drop to zero, whereas the second shows strange "jumps" to larger eigenvalues. We believe this corresponds to a failure to identify additional directions of variation, leading to a lower-rank kernel approximation than expected. For NeuralEF, this manifests as essentially repeating earlier eigenfunctions instead of finding new orthogonal directions.

The authors believe that one promising research direction for finding better self-supervised learning techniques would be to develop more stable or better-conditioned linear approximations of the positive-pair kernel, building on the spectral contrastive loss or NeuralEF. In particular, we see this as evidence that the parameterizations we used are not able to form good approximations of the true minimizer of the respective objectives. We hope that such techniques could be developed by combining ideas from the kernel methods and self-supervised learning communities, and that they would lead to useful representations for downstream supervised learning as suggested by our analysis.

### F.2 TRAINING DETAILS: OVERLAPPING REGIONS TASK

For each of the models visualized in Figure 6, we use a simple three-layer MLP with hidden layer sizes [64, 128, 256] which maps from the two-dimensional location of each grid point to the final kernel-dependent output embedding. We train all methods for 12,000 steps using a batch size of 1024 independently-sampled positive pairs per iteration, using the Adam optimizer (Kingma & Ba, 2014) with a cosine-decay learning rate schedule.

For the hypersphere kernel head $\widehat{K}_\theta(a_1, a_2) = \exp(h_\theta(a_1)^\top h_\theta(a_2)/\tau + b)$, we include a small regularization penalty encouraging $b$ to be small, which stabilizes training with the XEnt loss (since the loss is otherwise independent of $b$). When using the "XEnt + Logistic" loss mixture, we combine the two losses using a weight of 0.9 for XEnt and 0.1 for Logistic.

For all methods other than Neural EF, we used batch normalization for the first 6,000 iterations, then interpolated between the current batch statistics and the average from previous batches for 2,000 more iterations, and finally trained for 4,000 iterations using frozen batch norm statistics alone (e.g. in "inference" mode), which we found slightly improves eigenfunction quality. For Neural EF, we keep batch normalization at all steps, and in particular use L2 batch normalization for the output embedding as proposed by Deng et al. (2022).

To extract eigenfunction estimates from our kernel models, we compute estimates of the eigenfunctions and eigenvalues by performing population kernel PCA over the values of the kernel across all elements of $\mathcal{A}$, weighted by $p(a)$. For the NeuralEF model, we instead directly use the model's outputs as the eigenfunctions, and use running averages of $R_{ii}$ to approximate eigenvalues.

For Figure 6, we compute the alignment between eigenfunctions by taking their squared uncentered covariance $\mathbb{E}[f_i(a)\hat{f}_j(a)]^2$. Note that by Theorem 3.2 this is equivalent to the square of the coefficient

$c_i$ for the function $\hat{f}_j$ expanded in terms of the basis of eigenfunctions $f_i$; consequently we have $\sum_i \mathbb{E}[f_i(a)\hat{f}_j(a)]^2 = \mathbb{E}[\hat{f}_j(a)^2] = 1$. Since the specific choice of eigenfunctions is not uniquely determined when there are multiple eigenfunctions with the same eigenvalue, we sum the alignments for all eigenfunctions that have the same eigenvalue, leading to the block-diagonal structure shown in 6. We computed the positive-pair discrepancy for each approximate principal component function by analytically summing over all possible positive pairs.

**Population look-up table variant** In Figure 4, we use a modified procedure to improve visualization quality. Instead of using a MLP, we instead directly learn a lookup table of positions $v_a \in \mathbb{R}^2$ and scale modifiers $s_a \in [-5.0, 5.0]$ for each point $a \in \mathcal{A}$, and use a rational quadratic kernel head

$$\widehat{K}_\theta(a_1, a_2) = e^{s_{a_1}} e^{s_{a_2}} \left(1 - \frac{\|v_{a_1} - v_{a_2}\|^2}{2\alpha}\right)^{-\alpha}.$$

where $\alpha$ is a learnable parameter. (The size of each marker in Figure 4 represents the learned scale; we find that using the scale modifier improves eigenfunction quality, and note that more tightly-clustered points tend to have slightly smaller scales.)

We train using a full-batch variant of the XEnt and Logistic losses. For the XEnt loss, we compute

$$\mathcal{L}_{\text{XEnt}}(\theta) = -\sum_{a_1,\ a_2} p_+(a_1, a_2) \log p_\theta(a_2|a_1)$$

where

$$p_\theta(a_2|a_1) \propto p(a_2)\widehat{K}_\theta(a_1, a_2).$$

(Here weighting the kernel by $p(a_2)$ can be viewed as the population equivalent to sampling a set of negative pairs as the number of negative pairs approaches infinity.) For the logistic loss, we analytically compute

$$\mathcal{L}_{\text{Logistic}}(\theta) = \mathbb{E}_{p_+(a_1^+, a_2^+)} \left[-\log\sigma(\log \widehat{K}_\theta(a_1^+, a_2^+))\right] + \mathbb{E}_{p(a_1^-)p(a_2^-)} \left[-\log\sigma(-\log \widehat{K}_\theta(a_1^-, a_2^-))\right]$$

by summing over all possible positive and negative pairs. We use relative weights of $10\mathcal{L}_{\text{XEnt}} + 1\mathcal{L}_{\text{Logistic}}$.

We use population kernel PCA over the set $\mathcal{A}$ to identify the principal component functions. We then extend the principal component functions $f_i : \mathcal{A} \to \mathbb{R}$ across the full embedding space $h_i : \mathbb{R}^2 \to \mathbb{R}$ (ignoring the scale parameter for simplicity) according to

$$h_i(v) = \widehat{K}_\theta(v, \mathcal{A})^\top \widehat{K}_\theta(\mathcal{A}, \mathcal{A})^\dagger f_i(\mathcal{A}),$$

where $\widehat{K}_\theta(v, \mathcal{A})$ is the vector

$$\widehat{K}_\theta(v, A) = \begin{bmatrix} e^{s_{a_1}}\left(1 - \frac{\|v - v_{a_1}\|^2}{2\alpha}\right)^{-\alpha} \\ e^{s_{a_2}}\left(1 - \frac{\|v - v_{a_2}\|^2}{2\alpha}\right)^{-\alpha} \\ \vdots \\ e^{s_{a_{|\mathcal{A}|}}}\left(1 - \frac{\|v - v_{a_{|\mathcal{A}|}}\|^2}{2\alpha}\right)^{-\alpha} \end{bmatrix},$$

$\widehat{K}_\theta(\mathcal{A}, \mathcal{A})$ is the Gram matrix of the sequence $[a_1, \ldots, a_{|\mathcal{A}|}]$ (in other words, the matrix elements are defined by $[\widehat{K}_\theta(\mathcal{A}, \mathcal{A})]_{ij} = \widehat{K}_\theta(a_i, a_j)$), $\dagger$ denotes the Moore-Penrose pseudoinverse, and $f_i(\mathcal{A})$ is the vector $[f_i(a_1)\ f_i(a_2)\ \cdots\ f_i(a_{|\mathcal{A}|})]^\top$; this equation implicitly projects each point onto the appropriate principal component of the kernel. For numerical stability reasons, we regularize the pseudoinverse $\widehat{K}_\theta(\mathcal{A}, \mathcal{A})^\dagger$ by additionally removing eigenvalues very close to zero.

### F.3 TRAINING DETAILS: MNIST WITH MULTINOMIAL AUGMENTATIONS

Our goal in designing our task was to construct distributions $p(Z)$ and $p(A|Z)$ such that the exact positive-pair kernel could be computed, and so that the Markov chain would mix between different unperturbed dataset examples $z \in \mathcal{Z}$ without changing the label too frequently.

To this end, we constructed our task as follows:

- Define $\mathcal{Z}$ to be a particular subset of the MNIST dataset, and choose $p(Z)$ to be the uniform distribution over $\mathcal{Z}$. We consider two choices for $\mathcal{Z}$: randomly selecting 512 images from each of the ten digit classes (used for comparisions between models), and randomly selecting 1024 images from the digits 0, 1, and 2 (used for visualizations of the eigenfunctions).

- For each image, generate 64 pertubed copies, by randomly blurring, translating, and rotating digits by a small amount. Add a small constant to each pixel so that no pixel has value zero, then normalize each such copy so that its pixel values sum to 1.

- To generate an augmentation of an image $z \in \mathcal{Z}$ according to $p(A|Z = z)$, first choose one of the 64 copies of $z$, then sample a set of $k$ pixel locations with replacement from the distribution represented by that copy, where $k$ determines the augmentation strength. This means we are more likely to sample pixel locations which were within the original digit, although due to the perturbations described above the pixels may be scattered around the digit.

Our set $\mathcal{A}$ is thus the set of all $28 \times 28$ images for which all pixels have a nonnegative integer value, and the total of all pixels is $k$. (Due to the low pixel density, to improve visibility in our figures we render each pixel as a box 5x its original size, with shading indicating overlap of these boxes. However, when giving input to the model, we directly input this sparse pixel reprsentation, without the 5x multiplier.)

Given a particular augmented example $a$, we can compute $p(a|z)$ for any $z \in \mathcal{Z}$ by summing over each of the 64 copies of $z$ and using the closed-form PDF of a multinomial distribution. We can then compute $p(z|a)$ by normalizing over all possible $z$, and use this to exactly compute the positive-pair kernel feature map $\phi_+$. We selected the perturbation parameters such that there was nontrivial uncertainty in $z$ given each $a$; in other words, we made sure the positive pair Markov chain mixed well between examples.

### F.3.1 MNIST MODEL ARCHITECTURES

For the majority of our experiments, we used three-block variants of a ResNet-18 model followed by a two 128-dimensional fully-connected layers and a final output layer.

- Linear kernel: We used kernel parameterization $\widehat{K}_\theta(a_1, a_2) = h_\theta(a_1)^\top h_\theta(a_2)$ and the spectral loss, with $h_\theta$ as the output of the final layer. We set the dimension of the final layer to 512. We trained this model using the spectral contrastive loss.

- Hypersphere kernel, global scale: We used kernel parameterization $\exp(h_\theta(a_1)^\top h_\theta(a_2)/\tau + b)$, where $h_\theta$ is computed by normalizing the output of the final layer to lie on the unit hypersphere, and $b$ is a learned scalar. We set the dimension of the final layer to 32. We optimized the temperature $\tau$ and scale $b$ jointly with the model parameters. For the loss function, we used a weighted combination of 0.9 times the NT-XEnt loss and 0.1 times the NT-Logistic loss.

- Hypersphere kernel, individual scale: We used kernel parameterization $\exp(h_\theta(a_1)^\top h_\theta(a_2)/\tau + s_\theta(a_1) + s_\theta(a_2))$. We set the dimension of the final layer to 33, and defined $h_\theta$ by taking the first 32 entries and normalizing them to lie on the unit hypersphere. We then defined $s_\theta$ to be $5 \times \tanh(v_{33})$ where $v_{33}$ is the 33rd entry of the final layer. We again optimized the temperature $\tau$ jointly with the model parameters. The scale parameter allows the model to adjust the magnitude of the kernel on a per-example basis, which can be used to scale the eigenvalues of the principal components or to correct for differences in likelihood between points (since the magnitude of $K_+$ depends on the marginal likelihood of each point). For the loss function, we again used a weighted combination of 0.9 times the NT-XEnt loss and 0.1 times the NT-Logistic loss.

- Neural EF model: We set the dimension of the final layer to 512, and used L2 batch normalization on this final layer to ensure that the L2 norm of each output was 1 (e.g. that $C_j = 1$), as described by Deng et al. (2022). We used the modified version of the Neural EF objective described in Appendix E.2.

We trained our models using the Adam optimizer (Kingma & Ba, 2014) over 50,000 training iterations and a batch size of 4096 positive pairs per iteration, implemented using the JAX and FLAX libraries

(Bradbury et al., 2018; Heek et al., 2020). For methods other than Neural EF, we used batch normalization for the first 35,000 iterations, then interpolated between the current batch statistics and the average from previous batches for 2,000 more iterations, and finally trained for 13,000 iterations using frozen batch norm statistics alone (e.g. in "inference" mode); we found that this increased the quality of the extracted principal components. For Neural EF, we keep batch normalization at all steps due to the constraint that $C_j = 1$.

### F.3.2 EXTRACTION AND ANALYSIS OF PRINCIPAL COMPONENTS

To estimate the eigenfunctions of the true kernel, we performed PCA using the explicit form $\phi_+$ of the positive-pair kernel feature map, where the population covariance was approximated by averaging over 256 augmentations for each of the images in $\mathcal{Z}$. We then constructed the principal component projection functions using that covariance. Our alignment plots in Figure 7 for "True kernel" compare two PCA decompositions using independent random estimates of the population covariance.

For our approximate kernels $\widehat{K}_\theta$, we first constructed an approximation of the feature map for $\widehat{K}_\theta$ using the Nyström method (Williams & Seeger, 2000): we sampled a subset $S$ of augmentations by randomly selecting 25% of $\mathcal{Z}$ and sampling one augmentation for each image, computed the Gram matrix $\widehat{K}_\theta(S, S)$ for that subset, then set

$$\hat{\phi}(a) = \widehat{K}_\theta(S, S)^{-1/2} \widehat{K}_\theta(S, a)$$

where $\widehat{K}_\theta(S, a)$ is the vector of similarities of $a$ to each reference augmentation in $S$. The result is a feature map such that $\hat{\phi}(a_1)^\top \hat{\phi}(a_2) \approx \widehat{K}_\theta(a_1, a_2)$. We then again performed PCA using this feature map, using 256 samples per example in $\mathcal{Z}$ to compute the covariance.

For the Neural EF model, we again directly used the model's outputs as the eigenfunctions, and use running averages of $R_{ii}$ to approximate eigenvalues. We note that the Neural EF method did not find a fully orthogonal basis (as indicated by nonzero $R_{ij}$ terms for $i \neq j$), and some of its later eigenfunction estimates were correlated with earlier eigenfunctions; we did not attempt to correct for this in our plots in Figure 7. We believe this is the cause of the "jumps" from smaller eigenvalue approximations to larger eigenvalue approximations. (In contrast, the approximate eigenfunctions from the kernel PCA approaches are by construction uncorrelated over the sampled augmentations, due to being derived from eigenvectors of the sample covariance.)

We normalized all principal component projection functions to have unit uncentered variance, e.g. $\mathbb{E}[f_i(a)^2] = 1$ and $\mathbb{E}[\hat{f}_i(a)^2] = 1$. As for the overlapping regions task, we then computed alignments by taking the squared covariance $\mathbb{E}[f_i(a)\hat{f}_j(a)]^2$. We estimated the positive-pair discrepancy for each principal component function by taking the sample average of $(f_i(a_1) - f_i(a_2))^2$ over 16 randomly sampled augmentation pairs for each image in $\mathcal{Z}$.

### F.3.3 THREE-CLASS MNIST RATIONAL QUADRATIC MODEL

For Figure 1, we additionally trained a ResNet-18 model on only the MNIST digits 0, 1, and 2, using a scaled two-dimensional rational quadratic kernel:

$$\widehat{K}_\theta(a_1, a_2) = s_\theta(a_1)s_\theta(a_2)\left(1 - \frac{\|f_\theta(a_1) - f_\theta(a_2)\|^2}{2\alpha}\right)^{-\alpha}. \tag{10}$$

Here $f_\theta : \mathcal{A} \to \mathbb{R}^2$ embeds inputs into the plane, $s_\theta : \mathcal{A} \to \mathbb{R}^+$ is a learned scale function, and $\alpha$ is a learned shape parameter. We set the output dimension of the ResNet-18 model to 3, and took the first two elements as $f_\theta$; $s_\theta$ was defined as $\exp(5 \times \tanh(x))$ where $z$ is the third element. We also parameterize $\alpha = \exp(\gamma)$ and learn the scalar parameter $\gamma$. The model has a base hidden dimension of 128. The model was trained for 50,000 training iterations. We used the cross entropy InfoNCE loss (as described in Appendix B.1) and the Adam optimizer, with a batch size of 4096 positive pairs per iteration.

| | | Classification | | | Regression | | |
|---|---|---|---|---|---|---|---|
| # of sampled pixels $k =$ | | 10 | 20 | 50 | 10 | 20 | 50 |
| True Kernel ($K_+$) | Kernel PCA | 0.564 | 0.384 | 0.178 | 0.722 | 0.602 | 0.369 |
| Learned (Linear) | Kernel PCA | 0.589 | 0.398 | 0.254 | 0.724 | 0.603 | 0.362 |
| | ResNet Emb. | 0.553 | 0.375 | 0.229 | 0.730 | 0.567 | 0.459 |
| Learned (Hypersphere, Global Scale) | Kernel PCA | 0.594 | 0.401 | 0.421 | 0.722 | 0.559 | 0.660 |
| | ResNet Emb. | 0.561 | 0.371 | 0.229 | 0.737 | 0.602 | 0.518 |
| Learned (Hypersphere, Local Scale) | Kernel PCA | 0.583 | 0.392 | 0.276 | 0.718 | 0.552 | 0.524 |
| | ResNet Emb. | 0.575 | 0.380 | 0.206 | 0.742 | 0.599 | 0.522 |
| Learned (Neural EF) | Learned Eigfns. | 0.576 | 0.398 | 0.199 | 0.727 | 0.561 | 0.389 |
| | ResNet Emb. | 0.575 | 0.389 | 0.204 | 0.751 | 0.593 | 0.518 |

Table 2: Classification error (fraction misclassified) and regression error (squared error) on MNIST task with multinomial augmentations, across augmentation strengths $k = 10, k = 20, k = 50$.

## F.4 DOWNSTREAM SUPERVISED LEARNING ON MNIST

To better understand the performance of the true and approximate eigenfunctions for downstream supervised learning, we took our multinomial-sampling MNIST task, and compared the quality of various representations for two downstream prediction tasks: classification with a linear layer, and linear least-squares regression on the one-hot indicator vectors for each digit class. We considered three types of representation: the PCA projection functions for $K_+$, the PCA projection functions for each learned kernel $\widehat{K}_\theta$, and the intermediate layer embedding vector between the ResNet layers and the projection head for each model as proposed by Chen et al. (2020a). For each, we fit a regularized linear predictor on 160 labeled training examples (16 augmented samples from each class), using 160 additional validation examples to tune the regularization strength. For PCA representations, we additionally tune the representation dimension $d$, choosing only the first $d$ principal component functions.

The results are shown in Table 2. Performance is fairly similar across representations, suggesting that the positive-pair kernel $K_+$ captures much of the variability between augmentation strengths. Notably, at low augmentation strengths ($k = 50$), the true eigenfunctions have the smallest error, but eigenfunction approximations using Kernel PCA do not always outperform the intermediate layer representations, suggesting that inductive biases may play a role.

In more detail, to generate our supervised training and validation sets, we sampled one augmentation of 16 random images from each digit class, labeled with the original label, a total of 160 labeled augmentations in each set. For the test set, we took 170 distinct images from each digit class and generated one augmentation from each image, without overlap with the training or validation sets, for a total of 1700 labeled augmentations.

For the classification task, we fit a logistic regression classifier on the training set using SciKit Learn. For PCA representations, we swept over 40 logarithmically-spaced L2 regularization strengths from $10^{-4}$ to $10^1$, and also swept over representation dimension $d$, taking the first $d$ principal components for $d$ between 1 and 256. For ResNet embedding representations, we swept over 150 logarithmically-spaced L2 regularization strengths from $10^{-4}$ to $10^{10}$; we found that higher regularization strengths were necessary to attain a good solution. We selected the hyperparameters based on which setting gave the highest top-1 accuracy on the validation set.

For the regression task, we used a centered version of the one-hot indicator vector, e.g. the target vector for an example from digit 2 was

$$[-0.1, -0.1, 0.9, -0.1, -0.1, -0.1, -0.1, -0.1, -0.1, -0.1].$$

The purpose of this centering was to ensure that the expected value of the label vector was the zero vector. We then fit a predictor using ridge regression (L2-regularized least-squares regression) in Numpy. As for the classification task, for PCA representations we swept over 40 logarithmically-spaced L2 regularization strengths from $10^{-4}$ to $10^1$ and over each representation dimension $d$ between 1 and 256, and for ResNet embedding representations we swept over 150 L2 regularization

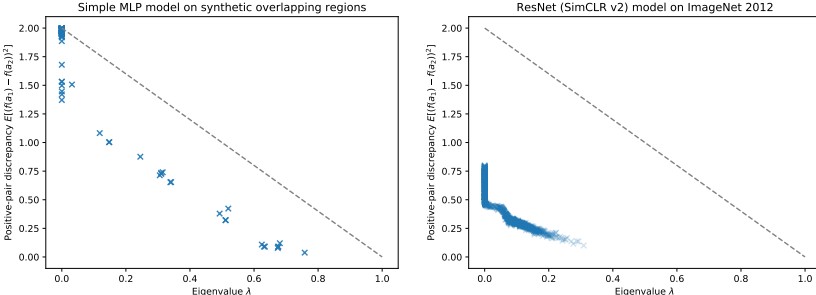

Figure 8: Relationship between kernel PCA eigenvalue and positive-pair discrepancy for a norm-constrained linear kernel head and the spectral contrastive loss, across two datasets. Relationship predicted by Equation 4 is shown with a dashed line. Eigenvalues smaller than $10^{-6}$ are omitted.

strengths from $10^{-4}$ to $10^{10}$. We selected the hyperparameters based on which setting gave the lowest squared error on the validation set.

### F.5 Principal Components Analysis of Spectral-Loss Linear-Kernel Model on ImageNet

In this section, we discuss some preliminary results from applying our Kernel PCA analysis to a real-world contrastive learning model.

We started by training a variant of the SimCLR v2 model Chen et al. (2020b) using the spectral contrastive loss and a linear kernel head, normalizing the output layer to have a constrained norm $\|h_\theta(a)\|^2 = c$ as described by HaoChen et al. (2021). We explored automatically learning this norm $c$ using the Adam optimizer and a separately-tuned learning rate. We found that the norm $c$ reliably increased during training, but training tended to destabilize and produce NaN weights once $c \approx 90$, and we were unable to successfully train a model with a higher output layer norm.

We trained a model for 100 epochs ($\approx$ 30,000 iterations) on the ImageNet 2012 dataset, using the default augmentation parameters and other hyperparameters for SimCLR v2, obtaining a similar supervised classification accuracy to previous results by HaoChen et al. (2021).

Next, we performed kernel PCA by using ordinary PCA with the explicit form of the kernel features (the output layer with normalization applied), since kernel PCA and ordinary PCA are equivalent for a linear kernel parameterization. We computed the covariance over a sample of 16 augmentations of each training dataset example, averaged across all examples. We then constructed the principal component functions $\hat{f}_i$ by normalizing based on the eigenvalues, and computed positive-pair discrepancies for each function over a sample of 8 augmentation pairs for each training dataset example.

Figure 8 shows the results of comparing the eigenvalues and positive-pair discrepancies, relative to the predicted relationship from Equation 4. For comparison, we also reproduce the corresponding figure for this kernel head parameterization and loss function on our overlapping regions task. We see that, across both tasks, the norm constraint causes estimated eigenvalues to be smaller than Equation 4 would predict, but there still appears to be an inverse correlation between the eigenvalue and positive-pair discrepancy. (On the overlapping regions task, it is close to a constant shift of the linear Equation 4 relationship. On the ImageNet task, the relationship is still somewhat linear, but with multiple irregularities, and a somewhat different slope than Equation 4 predicts; some of this may be due to increasing in the norm constraint during training.)

We also observe that, in both tasks, the sum of the eigenvalues from Kernel PCA is exactly equal to the norm constraint $c$. This suggests that the norm constraint is "capping" the sum of the eigenvalues, forcing the model to only learn a subset of eigenfunctions despite having capacity for more. We conjecture that stabilizing the learning dynamics might enable us to remove the norm constraint $c$ and thus capture additional eigenfunctions, leading to potentially superior representations for future self-supervised methods.

