# OpenReview forum: "Contrastive Learning Can Find An Optimal Basis For Approximately View-Invariant Functions"
_ICLR.cc/2023/Conference — ICLR 2023 poster_

### Official Review · Reviewer_7QsK · 2022-10-14

**Confidence:** 4
**Correctness:** 4
**Technical Novelty And Significance:** 3
**Empirical Novelty And Significance:** 4
**Recommendation:** 8

**Clarity, Quality, Novelty And Reproducibility:**

Clarity:
This paper is written very well. All the assumptions and theorems are clearly stated and lots of intuitions and explanations are provided. The limitations are also well discussed at the end.

Quality:
This is a solid theory paper that proves the minimax-optimality of contrastive learning methods by identifying their connection with Kernel PCA.

Novelty:
As far I know, this is the first paper that proves multiple contrastive learning methods can find a minimax-optimal representation for linear predictors assuming the target functions satisfy approximate view-invariance property.

Reproducibility:
I think both of theoretical analysis and the experimental results are reproducible.



**Strength And Weaknesses:**

Strengths:
This is a very solid and well-written theory paper that proves multiple contrastive learning methods can find a minimax-optimal representation for linear predictors assuming positive pairs have similar labels. This theory offers a unifying explanation for the success of multiple contrastive learning methods in practice. In the proof, they also build a very interesting connection between contrastive learning, Kernel PCA, and the Markov chain over positive pairs.

Weaknesses:
This paper builds a connection between contrastive learning and Kernel PCA, which is very interesting from a theoretical perspective. However, what's the implication of this theory? How does this knowledge further guide us to build better contrastive learning methods in practice?


**Summary Of The Paper:**

1. This paper proves that multiple contrastive learning methods can find a minimax-optimal representation for linear predictors assuming the target functions satisfy approximate view-invariance property (Assumption 1.1). To prove this, they first show that minimizing the contrastive loss is equivalent to solving Kernel PCA for the positive-pair kernel. Then they show that solving the Kernel PCA is equivalent to finding the eigenfunctions of the Markov chain over positive pairs, which allows them to prove the minimax-optimal property. They also give generalization bounds for the performance of the learned representations for downstream supervised learning.
2. In the experiments, in two synthetic datasets where the positive kernel has a closed form, the authors verified that multiple contrastive learning methods could indeed approximately recover the eigenfunctions of the kernel. The experiments also showed that the constraints on the kernel approximation and the weaker augmentation can have a negative impact on the recovery of the eigenfunctions.

**Summary Of The Review:**

This is a solid theory paper that proves the minimax-optimality of contrastive learning methods in finding representations for linear predictors assuming the view-invariance property of the target functions. The analysis also builds an interesting connection between contrastive learning, Kernel PCA, and the Markov chain over positive pairs. I believe this is a good step toward explaining the empirical success of various contrastive learning methods. My only concern is that it's unclear to me how this theory can guide us to design between contrastive learning methods.

---

> ### Author Response · Authors · 2022-11-15
> **Response to review**
>
> Thank you for your review, and we are glad you found our contributions in this work to be interesting and significant.
>
> One way that we hope our theoretical analysis will lead to better practical contrastive learning methods is by bridging the ideas behind contrastive learning and approximate kernel decomposition (using techniques such as SpIN and NeuralEF). In fact, as mentioned in our top-level comment, we recently encountered a preprint that builds on our submission by applying a new kernel decomposition technique to our positive-pair kernel; the result is a competitive practical self-supervised algorithm for real-world datasets, demonstrating that our theoretical results do have practical significance.

---

### Official Review · Reviewer_76h8 · 2022-10-24

**Confidence:** 3
**Correctness:** 3
**Technical Novelty And Significance:** 4
**Empirical Novelty And Significance:** 2
**Recommendation:** 6

**Clarity, Quality, Novelty And Reproducibility:**

Clarity:
- The paper is generally well-written except in my opinion in one part: at first sight, it is not obvious why this work relies on the Markov Chain over positive pairs and it would be useful to provide more intuition on this.

Quality:
- The theory seem sound and is validated by detailed experiments.

Novelty:
- The interpretation of contrastive SSL as kernel learning seems new to me.

Reproducibility:
- The code is not provided but Appendix contains enough experimental details.

**Strength And Weaknesses:**

Strength:
- This paper tackles understanding and unifying self-supervised learning methods, which is an important problem as this framework becomes increasingly popular and many apparently different methods are in fact doing the same thing.
- The theory is both simple and provides insights that follow the practice of SSL such as the importance of stronger augmentations or the relevance of linear evaluation, as well as new (at least to me) ones such as the negative effects of constraints on the output heads, and is validated experimentally.
- The main Assumption 1.1 is clearly stated and discussed.

Weakness:
- I think Assumption 1.1 does not apply to important downstream tasks for evaluating visual representations such as object detection and segmentation. Moreover, visual representations are evaluated via linear probing, but also via fine-tuning or knn, which is not covered here. Hence, the extent of this theoretical analysis seems at first sight limited. In my opinion it would be very interesting to discuss these limitations, as another possible valuable conclusion is that object detection / segmentation / fine-tuning / knn are not relevant when it comes to evaluate contrastively learned representations.

**Summary Of The Paper:**

This work claims that contrastive learning can find a minimax-optimal representation for linear predictors when the prediction function is approximately view-invariant.

More precisely, the authors demonstrate that learning a representation via contrastive losses such as NT-XEnt, NT-Logistic, and Spectral can be seen under some assumptions as learning a positive definite kernel between pairs of samples. Then, a finite-dimensional representation can be extracted via kernel PCA using the learned kernel. Under some assumptions, it is possible to demonstrate that the resulting representation is minimax optimal for linear prediction for approximately view-invariant functions. This demonstration is done by bridging the vectors obtained via kernel PCA to eigenfunctions of a Markov chain over positive pairs.

The authors then proceed to experimentally validate their theory on two synthetic tasks (overlapping regions and MNIST with specific augmentations).

**Summary Of The Review:**

This work offers a sensible kernel interpretation of contrastive self-supervised learning, whose understanding is an important problem today. The scope of this work may be limited since it differs from practical SSL in different ways:
- Models are not always linearly evaluated (knn and fine-tuning are also used).
- Augmentations distributions when learning downstream tasks often differs from training time.
- Important downstream tasks such as segmentation or object detection may not follow Assumption 1.1

I think these points could be at least discussed in the manuscript.

Overall, the pros outweight the cons and would tend to recommend acceptance.

---

> ### Author Response · Authors · 2022-11-15
> **Response to review**
>
> Thank you for your review, and we are glad you find our contributions novel and recognize the importance of understanding contrastive self-supervised learning.
>
> Regarding the Markov chain over positive pairs, we introduce the Markov chain because we believe this perspective leads to a particularly intuitive definition for the eigenfunctions of interest. In particular, functions satisfying Assumption 1.1 would be expected to approximately satisfy the Markov chain eigenfunction equation as well. We have added a sentence in Section 3.2 to clarify this intuition. On the other hand, the Markov chain is not strictly necessary, and it would be possible to instead recast section 3.2 in terms of a set of elementwise-rescaled eigenfunctions of the augmentation graph Laplacian discussed by HaoChen et al. (2021); we discuss this connection in more depth in Appendix C.3. Yet another alternative would be to frame the results in terms of the integral operator of the positive-pair kernel (Steinwart & Christmann 2008, Theorem 4.26), which is mathematically equivalent to averaging over Markov chain transitions. It is our opinion that these alternative definitions of the basis of interest are not as intuitive as the definition in terms of the Markov chain.
>
> We agree with the reviewer that the framework we introduce in Section 1 (including Assumption 1.1) does not cover all practical downstream applications of contrastive learning (such as object detection or segmentation), and that our optimality result does not apply to other types of downstream supervised learning such as fine-tuning or k-nearest-neighbors. We note, however, that our assumptions and downstream learning setup are in line with previous theoretical works studying contrastive learning, e.g. Wang & Isola (2020), Tosh et al. (2021), HaoChen et al. (2021), as well as with the common “linear evaluation protocol” used to evaluate self-supervised learning representations empirically, e.g. by Chen et al. (2020) for SimCLR and by Oord et al. (2018) for CPC. Additionally, we might still expect the general idea of view invariance, which our framework captures, to be useful in the other settings mentioned by the reviewer. If so, our theoretical results might partially explain the empirical good performance of contrastive learning representations for these other types of task and readout mechanism. We have added some additional discussion of the limitations of our analysis in Section 7.
>
> We hope our response addresses your concerns, and would be happy to provide additional clarifications if desired.
>
> —
>
> Steinwart, Ingo, and Andreas Christmann. *Support vector machines.* Springer Science & Business Media, 2008.

---

> > ### Comment · Reviewer_76h8 · 2022-12-09
> > **Acknowledgement**
> >
> > Thank you for your detailed answer. From your answer I conclude we both think the scope of this work may be limited compared to the common SSL practice. This is not a big concern to me, and I keep my score as is.

---

### Official Review · Reviewer_Ry9G · 2022-10-25

**Confidence:** 3
**Correctness:** 3
**Technical Novelty And Significance:** 3
**Empirical Novelty And Significance:** 2
**Recommendation:** 5

**Clarity, Quality, Novelty And Reproducibility:**

**Clarity**: the paper is in general clearly written. The major contribution should be highlighted more, though.

**Quality and Novelty**: the paper studies an interesting problem and the proposed viewpoint adds value to the existing theory.

**Reproducibility**: satisfactory.

**Strength And Weaknesses:**

**Strength**: the paper focuses on an important problem and the proposed viewpoint is interesting. The paper is in general well written.

**Weaknesses**: a few concepts as well as their connections are discussed in the paper. And (in addition) the notations are somewhat heavy. It would be helpful to have a few figures or tables for the readers to have a better understanding of the contribution of this paper.

**Summary Of The Paper:**

In this paper, the authors proposed a unified framework for interpreting a few existing contrastive learning methods as positive-pair (see Definition 2.1) kernel learning problems, see Table 1 for a summary.
Under a mild assumption for the target function g in Assumption 1.1, the authors showed:
* in Theorem 3.2 that standard kernel PCA under the proposed positive-pair kernel has a few interesting theoretical properties (to identify eigenfunctions of some Markov transition of interest); and
* in Theorem 4.1 that the obtained representation (1) maximizes the view invariance of the least-invariant unit-norm predictor and (2) minimizes the (quadratic) approximation error for the worst-case target function.

Some (somewhat toy) numerical experiments were provided in Section 6 to support the theoretical results.

**Summary Of The Review:**

I think the paper is in general interesting. It would be great if the authors could (1) find some way to better illustrate the contribution of this work and (2) address the following detailed comments:

* more discussion on the positive-pair kernel is needed: the probability ratio and the positive-pair kernel seem to be one key point in the paper, and it is a Mercer kernel. Do we know anything more about it? can we have some more insight into the task, etc.?
* it remains unclear to me how the proposed theory is of interest from a practical standpoint. Can that be used to design some novel methods/objectives beyond those listed in Table 1?

---

> ### Author Response · Authors · 2022-11-15
> **Response to review**
>
> Thank you for your review, and we are glad you found our viewpoint interesting.
>
> The reviewer notes that our paper discusses multiple concepts along with connections between them. We see these connections themselves as one of the main contributions of this work: our perspective unifies a number of concepts that have been touched on in previous work and shows how they all relate to each other. We have reworded the introduction to make our contributions more clear.
>
> We would like to point out that Sections 3.1 and 3.3 focus on discussing the properties of the positive-pair kernel that are important for our theoretical results, in particular that Kernel PCA with the positive-pair kernel produces the eigenfunctions that constitute the optimal representation. Moreover, the positive pair kernel is uniquely determined by the set of eigenfunctions discussed in Section 3.2, which follows from our derivation in Appendix C (and we have added a few more details about this fact there). We also describe the task we are considering in Section 1: we are interested in learning a representation that yields good performance when used for linear classification on a downstream supervised learning task.
>
> Although the focus of our paper is primarily theoretical, we believe that the connections between contrastive learning and kernel decomposition will enable the development of new practically useful training objectives. Indeed, our experiments already include a variant of Neural EF, a kernel decomposition technique that is not in Table 1 and was not originally designed for contrastive learning, and we show that this technique also enables us to extract the eigenfunction representation with comparable accuracy to the techniques from Table 1. Appendix E contains a more detailed discussion of how to use neural kernel decomposition techniques to build new objectives for self-supervised learning, which we were unable to include in the main body of the paper due to space limitations. (As mentioned in our top-level comment, we also recently discovered a recent preprint that builds on an earlier version of this work and uses our kernel definition to design a competitive practical objective for real-world datasets.)
>
> We hope this clarifies the contributions of our work, and would be happy to answer any additional questions the reviewer may have.

---

### Author Response · Authors · 2022-11-15
**Summary and updated manuscript**

We would like to thank all of the reviewers for their feedback on our manuscript. All of the reviewers highlighted the importance of the problem, and found our perspective to be well written, novel, and interesting.

In response to questions about the practical significance of our findings, we emphasize that our work has a primarily theoretical focus. Nevertheless, we believe that our contributions open up a number of avenues for future research toward practical self-supervised learning algorithms. One particularly exciting practical insight provided by our work is that kernel decomposition algorithms can be combined with our kernel to build well-motivated and theoretically-justified self-supervised objectives. (On this note, we recently encountered a preprint that builds on an earlier version of our submission by applying a new kernel decomposition technique to our positive-pair kernel, with strong results on real-world datasets, demonstrating that our theoretical insights do indeed have practical significance for the community. We are not directly linking to this preprint here, since the citation in that work deanonymizes our current submission, but we would be happy to give more details to the AC if this information would be useful.) We are also hopeful that our work inspires additional theoretical study of contrastive learning approaches under different assumptions or different types of downstream task, which could lead to additional practical insights.

We have revised the manuscript to address many of the specific comments made by the reviewers, including a more explicit discussion of contributions, some clarifications regarding the intuition for the Markov chain, some additional discussion of the limitations of our analysis, and additional details in the appendix. We have also added citations of some additional related work and made minor formatting changes to stay within the page limit. To make identifying the changes easier, new content is shown in green (and minor edits are shown in dark blue).

---

### Decision · Program_Chairs · 2023-01-20

**Decision:**

Accept: poster

**Justification For Why Not Higher Score:**

This is a solid, well-written paper, which helps to unify existing method. It deserves to be accepted. At the same time, as the reviewers note, there are certain limitations to the direct practical implications of the results.

**Justification For Why Not Lower Score:**

The paper provides novel insights into the relationship between popular existing contrastive learning methods, properties of the underlying positive pair kernel, and their applicability for invariant learning.

**Metareview: Summary, Strengths And Weaknesses:**

The paper performs a theoretical analysis of contrastive learning. It argues that a number of popular contrastive learning methods can be unified under the framework of kernel learning methods, learning different approximations to a certain “positive pair kernel”. The paper studies the performance of these methods ``approximate view invariance’’ assumption, in which the (unknown) target function g for the downstream task is near-invariant under the augmentation operation used in contrastive learning. The paper argues that kernel PCA on the positive pair kernel identifies eigenfunctions of a certain positive pair Markov chain, and uses this equivalence to prove that contrastive learning methods are minimax optimal for learning view-invariant functions, in the sense that out of all subspaces of a given dimension, the subspace of eigenfunctions minimizes the worst-case approximation error over view-invariant functions. It also provides generalization bounds for learning over this subspace.

Reviewers expressed a generally positive evaluation of the paper, noting that contrastive learning is a popular tool in practice, and that the paper helps to unify seemingly disparate methods — showing that they are all approximating a positive pair kernel. It provides insights into the structure of the positive pair kernel — showing that its eigenfunctions are eigenfunctions of a corresponding Markov chain, and are ordered by view-invariance. While similar objects have arisen in previous analyses of contrastive learning (esp the paper of Hao Chen et al), the submission conveys new insights — unifying a number of contrastive learning methods and proving minimax optimality of this eigenspace for invariant learning. The paper is very well written: both intuitive to read and mathematically crisp. As noted by the reviewers, the theoretical results of the paper pertain to a relatively clean application of contrastive learning (learn features, and then perform linear prediction in terms of kernel eigenfunctions), which does not cover all of the ways in which contrastive learning can be deployed in systems. Nevertheless, the paper conveys both qualitative insights and rigorous theoretical results regarding the relationship between popular contrastive learning methods, and their applicability to invariant learning.

**Note From Pc:**

if the above contains the word "oral" or "spotlight" please see: "oral" presentation means -> notable-top-5% and "spotlight" means -> notable-top-25%. As stated in our emails, we are disassociating presentation type from AC recommendations